# Transfer of sulfur and chalcophile metals via sulfide-volatile compound drops in the Christiana-Santorini-Kolumbo volcanic field

Clifford Georges Charles Patten [1,2] ✉, Simon Hector [2], Stephanos Kilias [3], Marc Ulrich [4], Alexandre Peillod[2], Aratz Beranoaguirre[2,5], Paraskevi Nomikou [3], Elisabeth Eiche[2,6] & Jochen Kolb [2]

Efficient transfer of S and chalcophile metals through the Earth's crust in arc systems is paramount for the formation of large magmatic-hydrothermal ore deposits. The formation of sulfide-volatile compound drops has been recognized as a potential key mechanism for such transfer but their fate during dynamic arc magmatism remains cryptic. Combining elemental mapping and in-situ mineral analyzes we reconstruct the evolution of compound drops in the active Christiana-Santorini-Kolumbo volcanic field. The observed compound drops are micrometric sulfide blebs associated with vesicles trapped within silicate phenocrysts. The compound drops accumulate and coalesce at mafic-felsic melt interfaces where larger sulfide ovoids form. These ovoids are subsequently oxidized to magnetite during sulfide-volatile interaction. Comparison of metal concentrations between the sulfide phases and magnetite allows for determination of element mobility during oxidation. The formation and evolution of compound drops may be an efficient mechanism for transferring S and chalcophile metals into shallow magmatic-hydrothermal arc systems.

Effective fluxes of sulfur (S) and chalcophile metals (e.g. Cu, Ag, Te, Au, Bi) in the Earth's crust at arc settings are key for the formation of large hydrothermal ore deposits (porphyry, epithermal, volcanogenic massive sulfide, skarns[1–3]). These fluxes can also lead to important S and metal emissions into the atmosphere, potentially impacting the Earth's climate[4–7]. The processes controlling such fluxes, however, are complex, with competitive effects during arc magmatic-hydrothermal evolution. Sulfur and chalcophile metals can be trapped in the lower crust via sulfide segregation[8,9] or can be released into the upper crust via magmatic volatile degassing[3,10–12]. The relative timing between such processes is critical in determining the fate of S and chalcophile metals[12–15]. The formation of sulfide-volatile compound drops (i.e.

droplets of sulfide melt attached to magmatic volatile phases) during magmatic evolution appears as an alternative and efficient mechanism for the transport of both S and chalcophile metals to the shallow crust[16]. Although such compound drops have been shown experimentally[16–19] and identified in magmatic ultramafic systems[7,20], their preservation in complex magmatic-hydrothermal arc environments is scarce[21,22]. Magmatism in arc environments is dynamic, involving mixing of variably evolved melts via hybridization or mingling. Injection of fertile mafic melt into differentiated magmatic chambers appears as a key process for transferring S and chalcophile metals to shallower magmatic-hydrothermal systems[5,23]. The evolution of the compound drops in such dynamic environments is, however,

[1]Institute of Mineralogy and Petrography, University of Innsbruck, Innsbruck, Austria. [2]Chair for Geochemistry and Economic Geology, Institute of Applied Geosciences (AGW), Karlsruhe Institute of Technology, Karlsruhe, Germany. [3]Department of Geology and Geoenvironment, National and Kapodistrian University of Athens, Athens, Greece. [4]Institut Terre et Environnement de Strasbourg, Université de Strasbourg, CNRS, Strasbourg, France. [5]Institut für Geowissenschaften, Goethe-Universität Frankfurt, Frankfurt, Germany. [6]Laboratory of Environment and Raw Materials Analysis, AGW, Karlsruhe Institute of Technology, Karlsruhe, Germany. ✉e-mail: clifford.patten@uibk.ac.at

poorly known and their role in the formation of arc-related hydrothermal ore deposits, although stipulated[16,21,22], remains cryptic.

In this study, we document the presence of compound drop remnants in the active magmatic-hydrothermal system of the Christiana–Santorini–Kolumbo (CSK) volcanic field within the 5 Ma-to-present Hellenic volcanic arc (HVA), Greece. Based on detailed petrography, high-resolution X-ray fluorescence mapping and in-situ mineral chemistry analysis we highlight the complex mineral reactions and elemental transfers which occur in compound drops during complex magmatic evolution. The findings highlight the role of sulfide-volatile compound drops in transferring metals to shallow magmatic-hydrothermal systems where ore deposits can form such as the hybrid epithermal-volcanogenic massive sulfide (VMS) mineralization at the Kolumbo volcano.

## Results
### Geological setting
The Christiana–Santorini–Kolumbo volcanic field is part of the HVA and formed in NE-SW oriented extensional basins within the Santorini-Amorgos tectonic zone[24]. The Santorini volcano, including Kameni, is the largest young volcanic centre in the HVA and records a complex volcanic history. After the Minoan eruption (ca. 3600 years ago) the volcanic islands of Palea Kameni and Nea Kameni formed in the centre of the caldera[25]. The Kameni islands show intermittent volcanic activity since the last 2200 years, producing dacite with little compositional variation[26]. The Kolumbo submarine volcano, located north-east of Santorini in the Anhydros basin, last erupted in 1650 CE[27]. Five distinct volcanic units are identified by seismic imaging (K1–K5). The composition of the volcanic products ranges from basaltic to rhyolitic lava flows and rhyolitic pumices[28]. Despite the proximity between the Kolumbo volcano and the Santorini and Kameni islands along the same tectonic line, seismic tomography and geochemistry suggest that both volcanic centres have distinct magmatic systems[29] tapping into different mantle sources[28].

### Sulfide-volatile compound drop remnants in the CSK volcanic field
The Christiana–Santorini–Kolumbo volcanic field is an active and complex magmatic-hydrothermal system[24] showing evidence of magma mixing at both the Nea Kameni and Kolumbo volcanoes. Sulfides associated with volatile phases have been observed in five andesitic enclaves from the 1939–1940, 1940–1941 and 1950 lava flows of Nea Kameni and within three pumice and lava samples from the K2 and K5 unit of Kolumbo (Supplementary Data 1).

Two sulfide populations related to volatile phases have been observed at Nea Kameni and Kolumbo volcanoes. The first population is defined by micrometric sulfide blebs associated with micrometric vesicles hosted by pyroxene phenocrysts from andesitic enclaves in porous dacite from Nea Kameni (Fig. 1c). The sulfide blebs are mainly spherical and contain pyrrhotite and chalcopyrite with minor magnetite (Fig. 1c), characteristic of magmatic sulfide blebs[30,31]. The vesicles associated with the sulfides vary in size from a few microns up to tens of microns (Fig. 1c). They are either directly in contact with or closely related to the sulfides. The sulfide-vesicle pairs have similar shapes, sizes, and mineralogy as those observed at the Merapi volcano[21], analogous to compound drops[16] and, hence, are interpreted as well as compound drop remnants. Locally, where the phenocrysts are fractured, the sulfide blebs are replaced by magnetite, hematite and minor covellite (Supplementary Fig. 2) most likely due to dissolution and oxidation by magmatic volatiles during magmatic decompression[21].

The second sulfide population observed is characterized by variably oxidized sulfide ovoids up to a few millimeters in diameter. They have a spherical-to-elongated ellipsoidal shape, are related to large millimetric vesicles (Fig. 1a, b, d–i), and occur at the interface between andesitic enclaves and dacitic host rocks at Nea Kameni and Kolumbo

(Fig. 1a, b, e, f) except for one ovoid from Kolumbo which is present within a rhyolitic matrix (Fig. 1i). The sulfide ovoids are either embayed in, in contact with or disconnected from the vesicles; the latter ranges in size from several hundreds of microns to a few millimeters (Fig. 1a, e, h, i, Supplementary Fig. 3e). The sulfide ovoids are constituted mainly of pyrrhotite and pyrite with minor pentlandite, chalcopyrite and covellite. Pyrrhotite appears as large grains (up to a few millimeters) with discrete exsolution of pentlandite. Pyrite occurs as a replacement of pyrrhotite generally along fractures filled with magnetite; additionally, pyrite proportion increases towards magnetite-rich zones (Fig. 1d). Pyrite shows cleavage-like texture and has relatively low reflectance. Chalcopyrite is scarce and has been observed in partly oxidized areas (Fig. 1d). It is often replaced by covellite associated with pyrite. Sulfides are replaced by fine-grained micrometric subhedral magnetite with discrete sulfide grains preserved in between (Fig. 1a, d, g). Sulfide replacement by magnetite ranges from limited, with discrete magnetite along fractures and at vesicle margins, to extensive, with almost magnetite-pure aggregates with discrete sulfides and porosity between the grains (Fig. 1d, g, Supplementary Fig. 3). Locally, magnetite is associated with hematite and shows a frothy texture (Supplementary Fig. 4). The sulfide ovoids, with their mineralogy and variably oxidized state, are similar to magmatic sulfides observed in different arc volcanoes which have sustained advanced oxidation during sulfide-magmatic volatile interaction[32].

### Sulfide and oxide mineral compositions
In-situ analysis of sulfide and oxide phases from the sulfide blebs and ovoids allows the characterization of their origin and a better understanding of metal behavior during compound drop evolution (Supplementary Data 2). Fresh sulfide blebs of compound drops in silicate phenocrysts show a distinctive magmatic signature characterized by relatively high chalcophile element concentrations (Co, Ni, Cu, and Ag; Fig. 2a) and an overall similar metal enrichment to compound drops from the Merapi volcano[21], and to sulfide droplets from mid-ocean ridge basalt (MORB)[31,33] and from arc volcanic rocks[34–36]. These similarities imply that the sulfide blebs formed from a sulfide-saturated melt. The sulfide blebs from the CSK volcanic field are more enriched in weakly chalcophile elements (Zn, As, Sb, Pb; Fig. 2a) compared to MORB sulfide droplets, which are enriched in strongly chalcophile elements (Co, Ni, Cu, Te, Au), supporting that sulfide saturation occurred in an andesitic melt, slightly more evolved than a MORB.

Pyrrhotite, pyrite and partly oxidized chalcopyrite from the sulfide ovoids show comparable metal enrichment to the sulfide blebs (Fig. 2b; Supplementary Discussion). The partly oxidized chalcopyrite is enriched in As, Ag, Au, Tl, Pb, and Bi relative to pyrrhotite and pyrite, while pyrite is enriched in Co, Ni, Mo, Sb, and Te (Fig. 2b, Supplementary Data 2). Such metal fractionation is akin to metal distribution during magmatic sulfide crystallization into monosulfide solid solution (MSS) and intermediate solid solution (ISS[33]) suggesting that the sulfide ovoids are well-differentiated magmatic sulfides overprinted by oxidation reactions.

Magnetite present within the sulfide ovoids has low concentrations in Ti, V, Mn, Co, Ni, and Zn, which are usually enriched in magmatic magnetite crystallizing from a silicate melt[37] (Fig. 2b, Supplementary Fig. 4) and differ in concentrations from magnetite crystallizing out of a sulfide liquid[38] (Supplementary Fig. 4). Hence, the analyzed magnetite did not form through magmatic process but rather by oxidation and replacement of the magmatic sulfides, as supported by petrographic observations.

## Discussion
Although the Nea Kameni and Kolumbo volcanoes share distinct magmatic plumbing systems they are both characterized by shallow magmatic chambers (~4 and ~5 km deep, respectively) fed by mafic melts from deeper sources[39–43]. For instance, the 2011–2012 unrest at

Santorini was due to the intrusion, into the shallow plumbing system, of new volatile-rich and more primitive mafic andesitic-basaltic magma[44]. The presence of magmatic sulfide blebs within the andesitic enclaves as well as the high S concentrations in andesite-hosted melt inclusions from Nea Kameni (~900–1000 µg g$^{-1}$ [45]) imply that the andesitic melt was sulfide-saturated upon injection into the shallow magmatic chamber. Additionally, the presence of trapped fluid

inclusions in olivine and pyroxene in andesitic enclaves at Nea Kameni[44] suggests that magmatic volatile exsolution occurred within a similar timeframe to sulfide saturation; allowing eventual volatile bubble nucleation on magmatic sulfide droplets and formation of sulfide-volatile compound drops (Fig. 3 [18]).

The similar mineralogy and composition, the spatial proximity as well as the close association with vesicles imply a genetic link between

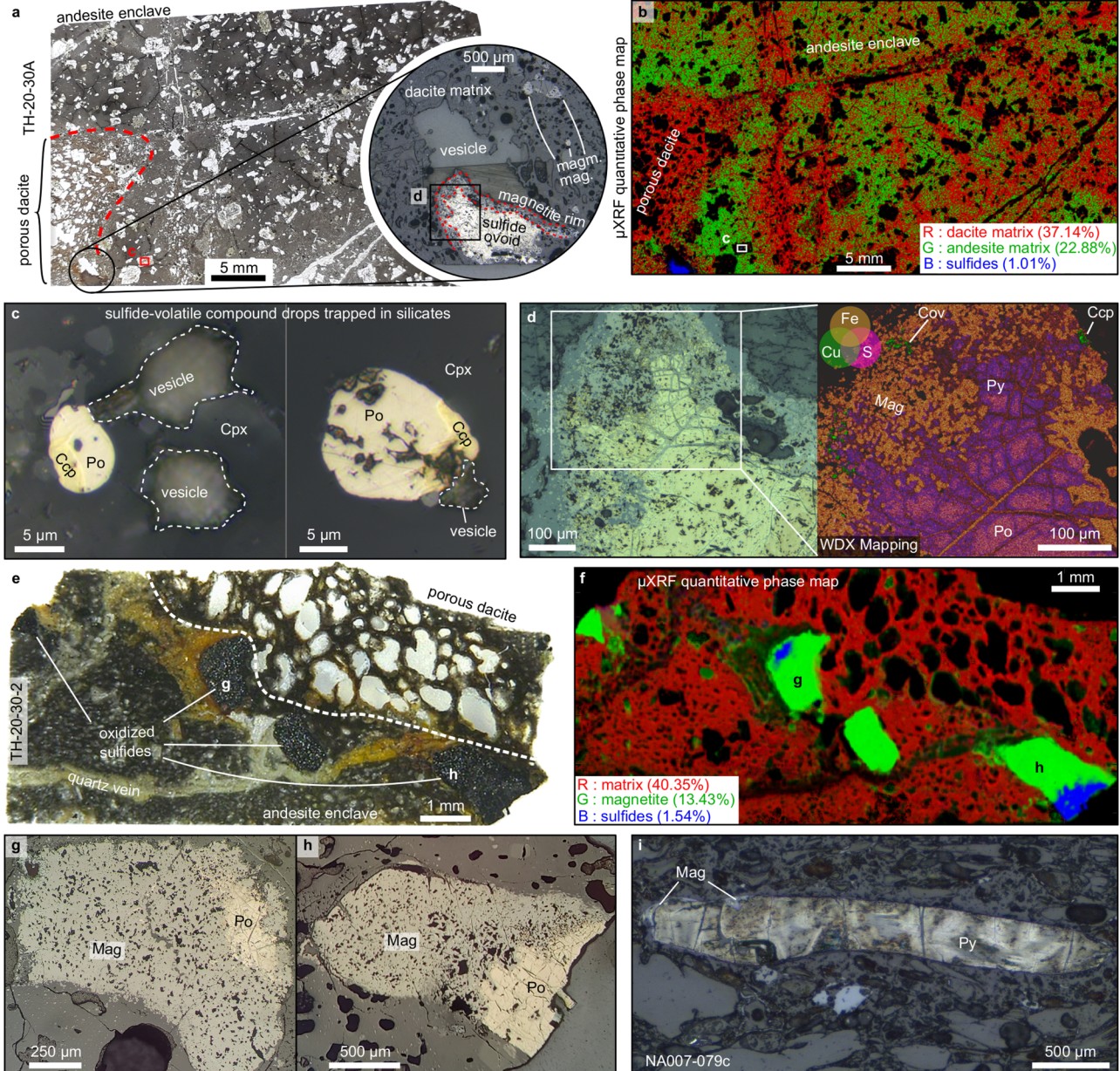

**Fig. 1 | Petrology of sulfide and oxide phases related to sulfide-volatile compound drops at Nea Kameni and Kolumbo volcanoes.** Petrology of sulfide and oxide phases related to sulfide-volatile compound drops at Nea Kameni (**a**–**h**) and Kolumbo (**i**) volcanoes. **a** Thin section of an andesite enclave in contact with porous dacite host rock from Kameni volcano. Sulfide ovoid associated with large vesicle at andesite/dacite margin (insert). **b** Micro-XRF quantitative phase map of (**a**) showing the proportion of dacite matrix, andesitic matrix and sulfide phases (pyrrhotite and chalcopyrite). Mineral proportions are calculated from elemental micro-XRF mapping. The map shows a complex interface between the andesitic enclave and the host dacite where sulfide-volatile compound drops occur. Black zones represent porosity and silicate phenocrysts. **c** Sulfide blebs and associated vesicles in pyroxene phenocryst. Sulfide blebs contain pyrrhotite (Po), chalcopyrite (Cpy) and magnetite (Mag). **d** Ovoid-shaped, partly oxidized sulfide melt occurring at the

dacite/andesite interface in (**a**). The sulfide ovoid is characterized by pyrrhotite (Po) in the core, pyrite (Py), chalcopyrite (Cpy), covellite (Cov) and magnetite (Mag) in the rim. The right part shows WDX scanning electron microscopy elemental mapping of S, Fe and Cu. **e** Thin section of an andesite enclave in contact with porous dacite host rock from Kameni also with partly oxidized remnants of sulfide-volatile compound drops. **f** micro-XRF quantitative phase map of (**e**) showing the proportion of silicate matrix (both dacitic and andesitic), hydrothermal magnetite and sulfide phases (pyrrhotite and chalcopyrite). Mineral proportions are calculated from elemental micro-XRF mapping. The map shows the relationship between the sulfides and replacing magnetite. **g, h** Details of pyrrhotite partly oxidized to magnetite. **i** Sulfide-volatile compound drop from the Kolumbo volcano within a rhyolitic matrix. The sulfide phase is characterized by pyrite with minor magnetite suggesting limited oxidation.

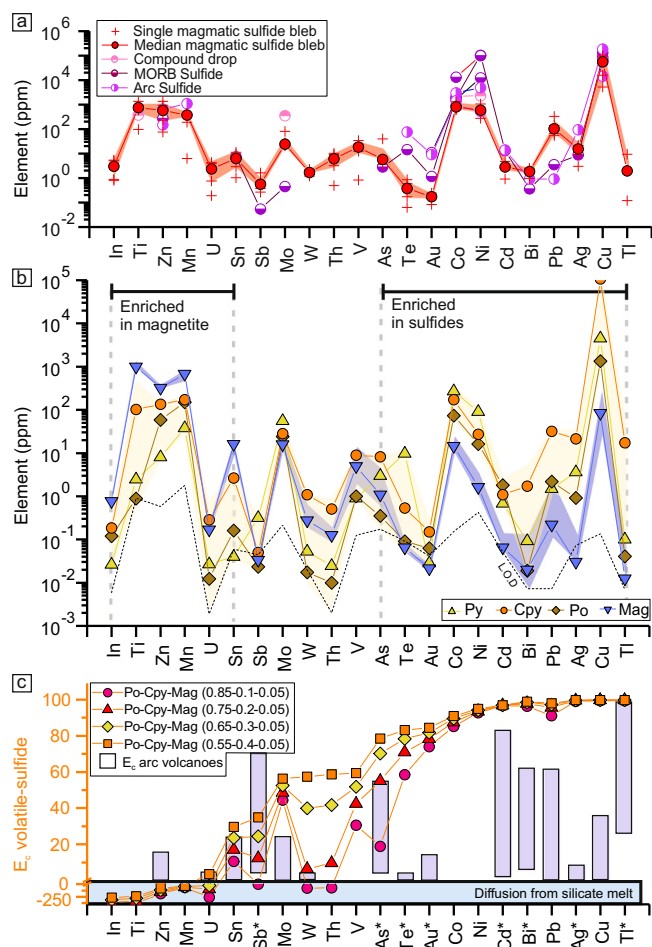

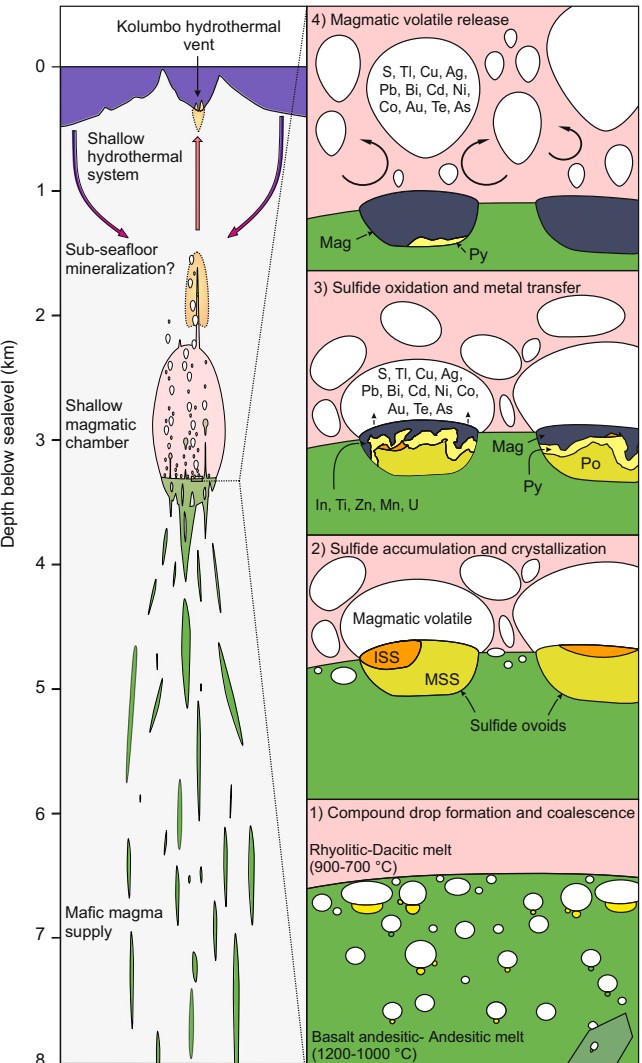

**Fig. 2 | Trace element concentrations in sulfides and magnetite related to compound drops as well as volatile-sulfide emanation and partition coefficients. a** Median trace element concentrations in magmatic sulfide bleb inclusions ($n = 4$). The shaded area corresponds to the upper and lower quartiles. Compound drops data are from ref. 21, MORB sulfides from refs. 31,33 and arc sulfides from refs. 34–36. **b** Median metal concentrations in pyrrhotite (Po), pyrite (Py), partly replaced chalcopyrite (Cpy) and magnetite formed after sulfide oxidation (Mag). The shaded areas correspond to the upper and lower quartiles. LOD limit of detection. **c** Calculated magmatic volatile-sulfide emanation coefficients ($E_c$). Arc volcano emission coefficients are from ref. 6. *Minimum emanation coefficient values calculated due to magnetite concentrations close to or below the LOD (see Supplementary Discussion).

**Fig. 3 | Model for S and chalcophile metal transfer during magmatic mixing.** Volcanic architecture based on the Kolumbo volcano[67,68]. 1) Compound drops formation and coalescence upon andesitic melt injection in the magmatic chamber. 2) Accumulation of compound drops in sulfide ovoids at the andesitic-dacitic melt interface. The sulfide liquid crystalizes into MSS and ISS. 3) Sulfide phases are oxidized into magnetite leading to S and chalcophile metal enrichment in the magmatic volatiles. 4) S and metal-rich magmatic volatiles are then released into the shallower hydrothermal system.

the magmatic sulfide blebs and the larger sulfide ovoids (Supplementary Discussion). Unlike small magmatic sulfide blebs (tens of microns), the relatively large size of the sulfide ovoids (>1 mm), does not favor flotation of sulfides through the magmatic crystal mush via compound drops (sulfide phase in compound drops <~0.4 mm[18]). The formation of the sulfide ovoids is rather likely to have occurred during magma mixing. Repeated injections of volatile-rich mafic (andesitic/basaltic) melts at Nea Kameni and Kolumbo result in their accumulation at the bottom of the magmatic chambers without extensive mixing with the intermediate/felsic (dacitic/rhyolitic) melts[42,44,46]. Continuous volatile exsolution from the mafic melts can lead to the formation of an intermediate hybrid layer between the mafic and felsic magmas[47,48]. The intermediate hybrid layer can eventually be incorporated into the felsic magmas, forming mafic enclaves[48]. The andesitic enclaves observed at Kameni show diffuse transition with the dacitic host rock, as revealed by X-ray fluorescence (μXRF) mineral mapping (Fig. 1b) suggesting they are remnants of the intermediate hybrid layer. The mafic enclaves also locally host quartz veins which

are crosscut at the andesite-dacite interface by the dacitic host rocks (Fig. 1e) implying that veins are not related to late hydrothermal processes but are of magmatic origin. Locally these veins crosscut the oxide/sulfide ovoids and show magmatic magnetites aligned at their margins (Fig. 1e), suggesting a sub-solidus process. Similarly to late residual silicate melt which partially fills the volatile fraction of compound drops at Norilsk'[20], the quartz veins are interpreted as late residual products of the mafic melt differentiation; further supporting that the mafic enclaves are remnants of an intermediate hybrid layer from the underplating mafic melt.

The interface between such underplating mafic melt and an overlying felsic melt is often a place where volatile bubbles, which have exsolved from the mafic melt, accumulate[47–49]. Similarly, magnetite-volatile aggregates, which also form in mafic magma and share similar physical properties to sulfide-volatile compound drops, have been shown to gather as well at such interface[50]. Hence, it is inferred that compound drops that are formed within the mafic melt rise toward the interface with the felsic melt where they stagnate and accumulate due

to differences in physico-chemical conditions. During migration, stagnation, and accumulation, the compound drops eventually coalesce[17], resulting in the formation of large sulfide ovoids (Fig. 3). During the coalescence of compound drops into larger sulfide ovoids, the temperature difference between the basaltic/andesitic and dacitic/rhyolitic melts (-1000–1200 °C and 750–900 °C, respectively[39,45,51]) leads the sulfide liquid to crystallize as monosulfide solution (MSS, -1050–1100 °C) and intermediate solid solution (ISS, -850 °C[52]) accounting for the differentiated pyrrhotite and chalcopyrite texture preserved in partly oxidized sulfide ovoids (Fig. 1).

The trigger for sulfide oxidation may be related to several processes such as magmatic degassing, change in volatile composition due to magma mixing or late-stage hydrothermal alteration. Magmatic differentiation in arc systems generally leads to magnetite crystallization and to increasing reducing conditions[8]. The presence of magmatic magnetites (Supplementary Fig. 4) suggests that the redox is driven towards more reduced conditions during the magmatic differentiation of the Kameni and Kolumbo volcanoes. Hence, it appears unlikely that magma mixing with more evolved melts affects the redox of the magmatic volatile phase towards more oxidizing conditions causing sulfide oxidation. Alternatively, arc volcanos are dynamic systems where long-lasting hydrothermal fluid circulation occurs and late-stage hydrothermal fluid circulation can cause sulfide oxidation. Such a process, however, would have altered the host rock leading to significant silicate alteration, which is not observed in the samples (Fig. 1e). It is rather inferred that, shortly after sulfide differentiation, sulfide oxidation occurs by reaction with the magmatic volatiles[14], likely through fluid/volatile-induced interface-coupled dissolution–precipitation mechanisms[53], upon magmatic degassing. Replacement of MSS (pyrrhotite) and ISS (chalcopyrite) by pyrite, covellite, and magnetite can be described with the following mineral reaction equations[7,14,21,22] (Fig. 3):

$$4FeS_{(po)} + H_2O_{(fluid)} + 2.5O_{2(fluid)} = Fe_3O_{4(mag)} + FeS_{2(py)} \\ + H_2S_{(fluid)} + SO_{2(fluid)} \quad (1)$$

$$3FeS_{(po)} + 2H_2O_{(fluid)} + 2O_{2(fluid)} = Fe_3O_{4(mag)} + 2H_2S_{(fluid)} + SO_{2(fluid)} \quad (2)$$

$$3CuFeS_{2(cpy)} + 2H_2O_{(fluid)} + 2O_{2(fluid)} = Fe_3O_{4(mag)} + 3CuS_{(cov)} \\ + 2H_2S_{(fluid)} + SO_{2(fluid)} \quad (3)$$

$$3CuFeS_{2(cpy)} + 1.5H_2O_{(fluid)} + 4.25O_{2(fluid)} = Fe_3O_{4(mag)} + 3CuHS_{(fluid)} \\ + 3SO_{2(fluid)} \quad (4)$$

$$3CuFeS_{2(cpy)} + 3HCl_{(fluid)} + 13O_{2(fluid)} = Fe_3O_{4(mag)} + 3CuCl_{(fluid)} \\ + 1.5H_2S_{(fluid)} + 4.5SO_{2(fluid)} \quad (5)$$

The magmatic sulfides partial oxidation into an assemblage of magnetite, pyrite, and covellite or complete oxidation to solely magnetite depends on the fO$_2$, fS$_2$, fluid composition, and temperature[21]. During ISS (chalcopyrite) oxidation, Cu is released into the volatile phase as diverse S- and Cl-complexes[54–56] of which Eqs. (4) and (5) are simplified representations.

The variably oxidized ovoids are either embayed into large vesicles (Fig. 1a, Supplementary Fig. 3e), partially associated with vesicles (Fig. 1i) or disconnected from vesicle (Fig. 1e, Supplementary Fig. 3d). This diversity of sulfide/oxide ovoid-vesicle textures suggests that the coalesced sulfide-volatile compound drops might eventually become unstable. At the andesitic-dacitic magma interface, changes in surface tension, drop compound size, fluid, and melt dynamics likely affect the stability of the compound drops and eventually lead to the separation of the volatile phase from the sulfide/oxide phases[18]. Ultimately, the magmatic volatile phase will rise within the magmatic chambers as plume[47,48] or through volatile pathways[57] and cracks in the magma crystal framework, directly feeding shallower hydrothermal systems[49] (Fig. 3).

Sulfide oxidation by magmatic fluids has been recognized as a key process for mobilizing S and metals to the shallow parts of arc magmatic-hydrothermal systems[6,15,21,32,58]. The efficiency, however, of metal transfer from the magmatic sulfides to the volatiles remains poorly known. The metal concentrations of the sulfide ovoids and replacing magnetite allow to constrain element mobility and calculation of emanation coefficients[6,59]. An emanation coefficient is usually used to determine element behavior upon eruption of lavas and is defined as[6,59]:

$$\varepsilon_x = (C_i - C_f)/C_i \quad (6)$$

With $C_i$ being the concentration of the element $x$ in a melt and $C_f$ the concentration of the element in the posteruptive lava. The emanation coefficients, although developed for lavas, can be applied to the compound drop system as it is based on the initial concentration of a system (here the magmatic sulfides) and the final concentration of the system after "emanation" (here the magnetite). It is however necessary to account for mass variation between the sulfide phases and magnetite during oxidation. Considering that the formation of 1 mole of magnetite requires the oxidation of 3 moles of sulfides (either pyrrhotite or chalcopyrite from Eqs. 2, 3 and 5) the emanation coefficient can be modified to:

$$\varepsilon_{x\,sulf-mag} = \left( C_{x\,sulf} - \frac{M_{mag}}{3*M_{sulf}} * C_{x\,mag} \right) / C_{x\,sulf} \quad (7)$$

With $C_{xsulf}$ and $C_{xmag}$ being the concentration of the element $x$ in the primary magmatic sulfides and magnetite, respectively and M$_{sulf}$ and M$_{mag}$ the molar mass of the sulfides and magnetite, respectively (see Supplementary Discussion for details). Numerous elements, such as Ni, Cu, Ag, Cd, Te, Au, Tl, Pb, and Bi show high mobility into the volatile phase ($\varepsilon_x$ ~ 80–100 %) while V, As, Mo, Sn, Sb, W, and Th show lower mobility ($\varepsilon_x$ ~ 0–80 %, Fig. 2C). Some elements show wide range in emanation coefficient values (As, Sn, Sb, Te, W, and Th) due to the relative proportion of chalcopyrite within the calculated magmatic sulfide composition as they have low concentration in pyrrhotite but higher concentration in chalcopyrite (Supplementary Data 2). Hence the relative proportion of chalcopyrite to pyrrhotite in magmatic sulfide has an impact on certain element mobility during oxidation. Some poorly chalcophile elements, such as Ti, Zn, In, Mn, and U, have negative emanation coefficients because the sulfide-volatile compound drops are not closed systems and interact with the surrounding silicate melt. During sulfide oxidation Ti, Zn, In, Mn, and U are interpreted to partition from the melt into the newly formed magnetite for which they have moderate to strong affinity[37]. Such partitioning is likely to have little effect on the other more chalcophile elements. The emanation coefficients determined from sulfide oxidation are notably higher than those observed from lavas at arc volcanoes[6], except for Zn and Sb (Fig. 2c). Although sulfide-volatile compound drops have been suggested to play a role in volcanic emissions their specific signature cannot be clearly distinguished from the silicate melt-volatile phase signature[6]. The high emanation coefficients determined from sulfide and magnetite analyzes imply that oxidation of the sulfides during sulfide-volatile compound drop evolution is a very efficient mechanism for releasing moderately to strongly chalcophile elements into the magmatic volatile phase. This process is possibly more efficient than silicate melt-volatile phase interaction and may directly supply metals to ore deposits[16].

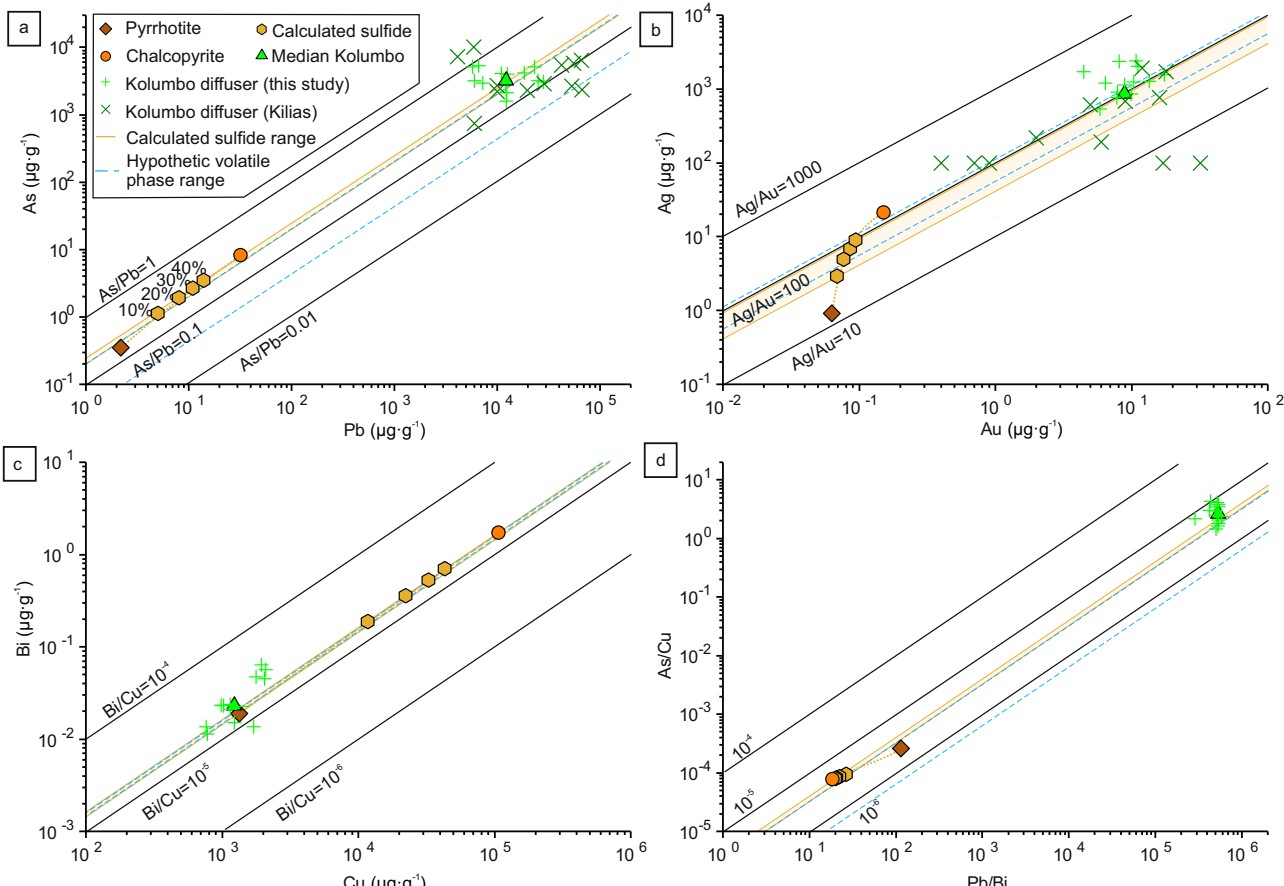

**Fig. 4 | Comparison of metal content in magmatic sulfides and mineralized samples from Kolumbo diffusers. a** Pb vs. As, **b** Au vs. Ag, **c** Cu vs. Bi and **d** Pb/Bi vs. As/Cu. The calculated magmatic sulfides are determined with various proportions of pyrrhotite and chalcopyrite (see Supplementary Discussion for details), the percentage represents the chalcopyrite proportion. The calculated sulfide range corresponds to the metal ratio of the calculated magmatic sulfides ranging from 10% to 40% chalcopyrite in proportion. The hypothetic volatile phase metal ratio range corresponds to the calculated sulfide ratio range factorized by the emanation coefficients of the selected elements (see Supplementary Data 3). The overlap between the magmatic sulfide, hypothetic volatile phase, and mineralization metal ratios highlights a genetic link between the compound drops and the mineralization. Kolumbo diffuser data are from ref. 60 and from this study (see Supplementary Data 6).

The Kolumbo volcano hosts an actively forming hybrid epithermal-seafloor massive sulfide (SMS) mineralization characterized by sulfide-sulfate diffusers enriched in As, Ag, Sb, Au, Tl, and Hg[60]. Metal endowment in Kolumbo is noticeably different from the metal enrichment in magmatic sulfides related to sulfide-volatile compound drops (Supplementary Fig. 5a). During magmatic volatile migration through the hydrothermal system of Kolumbo several processes such as fluid separation into brine and vapor phases, boiling and sulfide precipitation may occur; strongly impacting the metal load of the fluids and preventing direct comparison of metal content between magmatic sulfides and mineralization. To circumvent eventual metal fractionation processes, element pairs with similar behavior during hydrothermal fluid evolution and sulfide precipitation are used to test if a genetic link between sulfide-volatile compound drops and Kolumbo's mineralization could exist. This approach is similar to that of ref. 21 where metal ratios of magmatic sulfide and volcanic gases at Merapi volcano are compared to establish a genetic link as well as the modeling from ref. 15 which links compound drop and ore compositions at the Alumbrera Cu–Au deposit. Three element pairs are chosen: (1) As–Pb, representing elements that precipitate at low temperatures, (2) Cu–Bi, representing elements that precipitate at high temperatures, and (3) Au–Ag which are two elements with overall similar behavior[61–63]. For the three chosen pairs, magmatic sulfides and mineralized samples from the Kolumbo diffusers show good correlation and have similar As/Pb, Bi/Cu, and Ag/Au ratios (Fig. 4a–c) despite some natural variability in the diffusers. Although the volatile phase composition of the compound drops is unknown, the high emanation coefficients of the selected elements (over 90% for Pb, Ag, Cu, and Bi and up to ~80 % for Au and As, Supplementary Data 3) suggest little metal fractionation during sulfide oxidation. Hypothetic volatile phase metal ratios, defined as the calculated magmatic sulfide ratio of As/Pb, Bi/Cu, and Ag/Au factorized by their respective emanation coefficients, are used to assess metal fractionation during oxidation (Fig. 4). The hypothetic volatile phase ratio ranges show slight differences with the calculated magmatic sulfide ones (Fig. 4), with only the As/Pb showing a wider range as As has the lowest emanation coefficients of the selected elements, implying limited fractionation with Pb. Nevertheless the magmatic sulfide and the hypothetic volatile ratio ranges overlap well with that of the mineralization from Kolumbo (Fig. 4). Hence, the metal ratios of the magmatic sulfides are good proxies for the metal ratios of the volatile phase, allowing comparison with the mineralization (Fig. 4). Other element pairs also show similar distribution (Au/Tl, Tl/Ag, Te/Cu, Supplementary Fig. 5b, c). Comparison of the As/Cu and Pb/Bi ratios (Fig. 4d) allows to compare element pairs with opposite geochemical behavior and further supports the similar geochemical signature between the magmatic sulfides and mineralization despite the effect of metal fractionation (as well as Cu/Au and Bi/Ag ratios, Supplementary Fig. 5e). Hence, a genetic link

possibly exists between magmatic sulfides and the mineralization at Kolumbo, supporting that oxidation of sulfide-volatile compound drops can directly supply metals to ore deposits[16]. Finally, Cu, Te, and Bi have high emanation coefficients during sulfide oxidation (Fig. 2c) and show similar Cu/Bi and Cu/Te ratios between magmatic sulfides and diffusers, but these elements are poorly enriched in Kolumbo mineralization relative to other metals (Fig. 4 and Supplementary Fig. 5[60]). Fluids venting at the Kolumbo hydrothermal field have temperatures up to 265 °C[64] suggesting that significant Cu, Te, and Bi might be enriched below the diffusers at higher temperatures, possibly as a Cu-rich stockwork or deeper as a porphyry-style mineralization. The role of compound drops in the CSK volcanic field suggests that in addition to porphyry and magmatic Ni-Cu-PGE deposits compound drops can also be directly involved in the formation of epithermal and VMS deposits.

## Methods

### Micro-XRF mapping
Micro-XRF measurements were done on 30-μm-thick polished thin sections using a micro X-Ray fluorescence (μ-XRF) Brucker M4 Tornado at the Institut Terre et Environnement de Strasbourg (France). The instrument is equipped with a Rh anode operating at 400 μA with an accelerating voltage of 50 kV. Polycapillary lenses were used to focus the X-ray beam down to 20 μm full-width-at-half-maximum at the sample surface. Two energy-dispersive silicon drift detectors of 125 eV resolution, and with an active area of 60 mm² each were used to measure fluorescence spectra (180 s counting time per spectrum). Measurements were performed in a vacuum chamber at 2 mbar to minimize air absorption and ensure the best signal-to-noise ratio. μ-XRF chemical mappings were acquired with a 30 μm step interval and a dwell time of 700 ms/pixel. For each map, the color scale corresponds to the intensity of the Kα-lines of elements (Na, Mg, Al, Si, K, Ca, Cr, Ti, Mn, Fe) calculated from the integration of a specified region of interest (ROI) of the energy range of XRF spectra (Supplementary Figs. 6–9). Phase maps were calculated using the calculation procedure developed in ref. 65.

### In-situ mineral analysis
In-situ trace element analyzes of sulfide and oxide phases were carried out by Electron Micro Probe Analysis (EMPA) and by Laser Ablation-Inductively Coupled Plasma-Mass Spectrometry (LA-ICP-MS). Electron Micro Probe Analysis analyzes were carried out at the Institute of Geological Sciences from the University of Bern using a JEOL JXA-8200 Superprobe equipped with five wavelength dispersive crystal spectrometers (WDS) and one energy-dispersive spectrometer (EDS). A 15 kV acceleration voltage and a 20 nA probe current were used for analysis. Natural and synthetic oxides, sulfides, and silicate minerals were used as standards. LA-ICP-MS analysis was carried out at the Laboratory of Environment and Raw materials Analysis (LERA), Karlsruhe Institute of Technology, using a Teledyne 193 nm Excimer Laser coupled to an ICP-MS (Element XR ThermoFisher). Analyzes of pyrrhotite, pyrite, chalcopyrite, and magnetite were done with spot size of 35 μm, laser frequency of 10 Hz, fluence of 5 J cm⁻², and helium and nitrogen flow of 0.3 L.min⁻¹ and 10 mL.min⁻¹, respectively. The $^{57}$Fe from EMPA analyzes were used for internal standard calibration. The following isotopes were measured: $^{29}$Si, $^{32}$S, $^{49}$Ti, $^{51}$V, $^{55}$Mn, $^{57}$Fe, $^{59}$Co, $^{60}$Ni, $^{63}$Cu, $^{66}$Zn, $^{75}$As, $^{95}$Mo, $^{107}$Ag, $^{111}$Cd, $^{115}$In, $^{118}$Sn, $^{121}$Sb, $^{125}$Te, $^{182}$W, $^{197}$Au, $^{205}$Tl, $^{208}$Pb, $^{209}$Bi, $^{232}$Th, $^{238}$U. Calibration and data quality checking was done using the sulfide-pressed pellets Fe-S1, Fe-S4, and PTC1b from UQAC University for sulfide analyzes and basaltic glasses BHVO-2G, BCR-2G, and BIR-1G from the USGS for oxide analysis. Although non-matrix matching, calibration of magnetite using Fe-rich basaltic standards is adequate[38]. Data reduction was done using the Iolite software[66] 3DRS plugin in two separate runs for sulfide and oxide analyzes. Accuracy and precision for reference materials (<15% for most elements) as well as limits of detections are detailed in Supplementary Data 4.

### Diffuser whole rock metal concentration
Thirteen samples from a > 1 m high diffuser from the Politeia area in the Kolumbo hydrothermal field[60] have been analyzed for metal concentrations. Whole rock analysis was carried out by acid-digest-ICP-MS method using a Thermo X-series 2. Pulverized samples (100 mg) were dissolved in a combined $HNO_3$-HF-$HClO_4$. The powder was heated in a closed Teflon vessel for 16 h at 120 °C (DigiPrep, SCP Science) using 40% HF (suprapur), 65% $HClO_4$ (normatom) and 65% $HNO_3$ (sub-boiled). The final residue was dissolved in 50 mL of ultrapure water. The following isotope were selected for analysis: $^{55}$Mn, $^{57}$Fe, $^{59}$Co, $^{60}$Ni, $^{63}$Cu, $^{66}$Zn, $^{75}$As, $^{95}$Mo, $^{107}$Ag, $^{111}$Cd, $^{121}$Sb, $^{126}$Te, $^{182}$W, $^{197}$Au, $^{205}$Tl, $^{208}$Pb, $^{209}$Bi and $^{232}$Th. Calibration solutions were prepared with the CertiPUR® ICP Multi-Element Standard VI (Merck) used as stock solution and different single-element standards. To identify and correct measurement drift, 50 μL of an internal standard containing scandium ($^{45}$Sc), rhodium ($^{103}$Rh), indium ($^{115}$In), and thulium ($^{169}$Tm) were added to all samples, including the blanks (1% $HNO_3$). For all measurements the following Ar gas flows were chosen: Nebulizer gas, flow of 0.88 L/min, auxiliary gas flow of 0.68 L/min, and cooling gas flow of 13 L/min. Data quality was assessed by the analysis of the reference materials CH-4 and TUBAF (Supplementary Data 6).

## Data availability
Data used in this study are available in the Supplementary Data and Supplementary Information.

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

## Acknowledgements

This study is part of the project ASK, funded by the DFG SPP program DOME grant PA 3523/2-1 (C.G.C.P) and the DFG grant INST 121384/213-1 FUGG (J.K.). Samples from Kolumbo volcano were collected during the E/V Nautilus cruise in, 2011 supported by U.S. National Oceanic and Atmospheric Administration, Office of Ocean Exploration, and the Ocean Exploration Trust.

## Author contributions

C.G.C.P., S.K. and J.K. conceived the project, C.G.C.P., S.H., A.P., S.K. and P.N. participated in field sampling, P.N. participated to on-board sampling during the E/V Nautilus 2011 cruise, M.U. performed µXRF analyses, A.B. performed LA-ICP-MS analyses, E.E. assisted with laboratory support, C.G.C.P and S.H. wrote the manuscript, all authors participated in manuscript editing.

## Funding

## Competing interests

The authors declare no competing interests.
