## [Peer Review File · Nature Communications]

Editorial Note: Figures on page 45 of this Peer Review File have been redacted as indicated to remove third-party material where no permission to publish could be obtained.

REVIEWER COMMENTS

Reviewer #1 (Remarks to the Author):

Manuscript NCOMMS-22-45611-T “Sulfur and chalcophile metal transfer via sulfide-volatile compound drops during magma mixing: evidence from the Christiana-Santorini-Kolumbo volcanic field” by C.G.C Patten, S. Hector, S.P. Kilias, M. Ulrich, A. Peillod, A. Beranoaguirre, P. Nomikou, E. Eiche, J. Kolb.

This is a very interesting paper dealing with the transfer of sulfur and chalcophile elements from the magma to the fluid phase, via the formation of sulfide-volatile compound drops. The manuscript illustrates possible natural examples of compound drops occurring in andesitic magmas mixing with dacitic magmas, which were observed in the eruptive products of the Christiana-Santorini-Kolumbo volcanic system. The manuscript is well written and clear, the figures are of high quality. Given the topic and the quality of the paper, I recommend it for publication in Nature Communications, after moderate revisions.

I suggest the authors to improve the discussion of two points in order to strengthen the paper: (i) the timing of sulfide oxidation and the mechanism operating it, and (ii) the calculation and the interpretation of volatile-sulfide partition coefficients.

The timing of sulfide oxidation and the mechanism operating it are not fully elucidated in the present version of the manuscript. It is unclear whether the fluids forming the compound drops are the same operating sulfide oxidation. Figure 3 shows a continuum between stages 2 and 3, but the occurrence of pyrite replacement along fractures would rather suggest the existence of two different events. Why the sulfides are oxidized? As the manuscript is short, in my opinion there is room for discussing these points, on the basis of textural observations. Several possibilities could be presented, for instance:

- sulfide oxidation could be due to a change in volatile composition due to magma mixing (volatiles from the dacitic magma would need to be more oxidizing, why?);
- sulfide transformation into magnetite could occur as a consequence of desulfurization during degassing (as described in Iacono-Marziano et al. 2022);
- sulfide oxidation could occur in a later stage (what could be the origin of these late fluids?).

I am also concerned by the direct comparison between volatile-sulfide partition coefficients calculated in this study and vapor-melt and brine-melt partition coefficients, as shown in Fig. 2. I do not think this comparison is meaningful, and I would like to see a more convincing evidence of the efficiency of compound drops in transferring chalcophile elements into shallow magmatic-hydrothermal arc systems. I also suggest to explain why volatile-sulfide partition coefficients are calculated with the formula at line 187, this is unclear.

Minor comments:

Line 73: I would remove “Evidence ...is defined by”, and simply speak about two sulfide populations.

Line 75: Fig.1a is cited as illustrating the micrometric sulfide blebs, I do not consider it as the most appropriate.

Line 112: Fig. 2b, instead of Fig. 2a?

Fig. 2: please correct KD sulfide-volatile in the figure legend

Reviewer #2 (Remarks to the Author):

Review of manuscript NCOMMS-22-45611-T

Title: Sulfur and chalcophile metal transfer via sulfide-volatile compound drops during magma mixing: evidence from the Christiana-Santorini-Kolumbo volcanic field

By C.G.C. Patten¹, S. Hector¹, S. P. Kiliass², M. Ulrich³, A. Peillod¹, A. Beranoaguirre¹, P. Nomikou², E. Eiche^{1,4}, J. Kolb¹

Submitted to Nature Communications

In this manuscript, authors describe sulfide-oxide-vesicles associations observed within samples from the Christiana-Santorini-Kolumbo (CSK) volcanic field. These associations are interpreted as 'fresh' and oxidized 'compound drops'. The authors interpret larger millimetric sulfide ovoids as being the result of coalescence of smaller compound drops, and the oxides (magnetite) as being the result of oxidation of primary magmatic sulfides during sulfide-volatile interaction. They then study metal compositions of the sulfides and the oxide phases in order to evaluate magmatic volatile-sulfide partition coefficients. They finally use all these observations to infer the potential impact of 1) sulfide droplet – volatile association, 2) the coalescence of these compound drops and 3) the oxidation of these sulfides on S and metal transfer mechanism between the sulfides and shallow magmatic-hydrothermal fluids in arc systems and therefore ore genetic processes in arc systems. Finding evidence of such processes in volcanic rocks can be challenging, especially being able to quantify and observe the actual transfer of metals from sulfide liquid to magmatic-hydrothermal fluids, and this would be one of few studies doing so. These fluids would then have the potential to become mineralizing fluids for hydrothermal Cu-systems, which has significant implications on the genetic models for hydrothermal mineralized systems. However, I think the dataset presented here, in its actual form, does not provide strong enough evidence for such a claim. In order to evaluate whether it could, the authors would need to provide complementary information.

1. Quality of the data, Analytical approach, Data and methodology

The data presented seems to be of high quality with appropriate techniques and methods used. However, the authors need to present the data in its integrity. For example, sulfide composition data presented in supplementary material is for specific sulfides (pentlandite, pyrrhotite, chalcopyrite, etc...), and is only showing medians. To truly evaluate the quality of the manuscript we would need the whole dataset in supplementary material:

- all analyses collected as part of this study, not just median values, along with context (sulfide bleb vs ovoid, sulfide in inclusion?, etc....).
- Error bars (uncertainties on all the analyses) need to be added to be able to do the comparisons they are doing.
- More details on the methods need to be added in supplementary material (referring to Nadeau et al 2010's supplementary material is too complicated for reviewers and future readers). Such as How average sulfide bleb compositions were calculated using individual phases' composition – see Supp. Table 2.
- Better contextual information for the samples is needed, especially precise location (schematic section

through the system showing where the samples come from, map?): How many samples were studied as part of this work exactly? How representative of the overall system are they? Are these two 'populations' only described on the basis of 2-3 samples that have been described in detail here? If so, then this does not seem statistically representative enough. Are there more observations of these two sulfide populations in the system as a whole, within the samples, within the dataset?

- Overall, a detailed sample description and location information needs to be added. We currently cannot evaluate how representative the selected samples are of the system and whether or not the amount of observation and data collected is enough to be significant.

1. Level of support for the conclusions

There are many interpretations and conclusions present in this manuscript that need stronger support:

- Compound drops

The fact that what we are observing are indeed compound drops needs to be better proven (rather than stated). This manuscript needs to add a section demonstrating the fact that what they are describing as 'compound drops' are indeed preserved/frozen in place sulfide bleb - gas bubble pairs. Not just refer to supplementary material (that does not really demonstrate it either in the text). Not simply compare the sulfide blebs with sulfide blebs from other described compound drops in the literature, but actually describe and compare with literature compare the whole sulfide-gas bubble association/pair.

- Sulfide composition:

Not enough data to support the compositional comparison between the sulfides from this study with MORB sulfides and compound drops sulfides (See figure 2A). More data (from literature) needs to be added to support the comparison and conclusion.

As for the data collected as part of this study, we need more information as well: 1) How many sulfide bleb inclusions in plagioclase were analysed? 2) composition shown in the diagram is this the 'evaluated' composition of the whole bleb? How was that determined? Or is it the composition of specific sulfide phases within the bleb?

When comparing composition of sulfides in sulfide bleb with sulfides in ovoid, you need to compare the same thing, i.e. composition of individual phase type, or compare the calculate composition of the overall sulfides. This is currently not the case in figure 2.

- Evidence of oxidation process of the sulfides?

Can the authors use mineral maps and mineral compositions to do a rough mass balance for the oxidation process of the sulfides?

2. Potential significance of the results

Many of the processes and implications described in this paper have already been stated in previous papers (especially in Nadeau et al 2010, Mungall et al 2015, Edmonds and Mathers 2017, Edmonds et al 2018) and aren't novel as such. However, actual evidence in the rocks of the processes described here

(presence of compound drops in andesite enclaves, coalescence of sulfide-bubble compound and especially metal transfer into magmatic hydrothermal fluids through the oxidation process of the sulfides) would be very significant.

In conclusion, this is a manuscript that reads well and describe a very interesting process and its potential implications. The approach is valid and the data is generally well presented. However, in its current form, interpretations and conclusions are not well enough supported by the data (or at least not in the state in which it is provided).

I have also added numerous reviews, comments, and suggestions to improve this manuscript on the attached annotated versions of the files provided for the review.

Reviewer #3 (Remarks to the Author):

This is a relatively concise and potentially meaningful contribution to the discussion on the vertical transport of sulfur and metals in the magmatic plumbing system. The formation and dynamics of sulfide-vapour compound drop become one of the hot research topics in recent years since the groundbreaking work of Mungall, and here the authors provide high-resolution X-ray fluorescence mapping and in-situ mineral analysis of compound drops found in the Christiana-Santorini-Kolumbo volcanic field. The estimation of the magmatic volatile-sulfide partition coefficients of metal elements based on their concentrations within sulfide and secondary magnetite is a novel idea, but I must note that the calculation behind this idea seems to be too crude, or even not right if just reading their manuscript and supplementary information. Secondly, I guess this manuscript was transferred from other journal, and thus there are not so much information in the short manuscript, which weakens its reliability. The geological setting of CSK volcanic field was introduced in the supplementary information, but it just occupies half of the page and can be moved to the major manuscript. In addition, all information about the magma evolution and mixing of CSK volcanic field is a summary of previous articles, and no new idea, data and information is reported in the supplementary information. Hence, this manuscript is a little thin on content, especially given that their new idea about the magmatic volatile-sulfide partition coefficients is based on an overly simplistic calculation. Thirdly, the dynamics of compound drop during magma mixing and mingling should be the focus of this work, but the associated discussion and interpretation are likely qualitative conjecture, which may not deep our current understanding of this topic. Overall, this is an interesting and ordinary article about the sulfide-vapour compound drop, but it may have a long way to reach the high level of Nature Communications. My recommendation as of this reading is, narrowly, major revision, but would understand if the suggested revisions cannot be done in the time required for a revision and thus a rejection with resubmission is required, and will leave that to the judgment of the editor.

Major comments:

1. Model for the calculation of magmatic volatile-sulfide partition coefficients:

In the manuscript, “the magmatic volatile-sulfide partition coefficients are calculated as follows”:

$$K_{(D \text{ magma volatile/sulfide})} = 1 + (C_{\text{Sulfide}} - C_{\text{Magnetite}}) / (C_{\text{Magnetite}})$$

But the authors had not provided any theoretical explanation about this equation. They may assume that the metal concentration in magmatic volatile is equals to the difference between the sulfide and the replacing magnetite (i.e., $C_{\text{volatile}} = C_{\text{sulfide}} - C_{\text{magnetite}}$), but why this value should be divided by the metal concentration of magnetite ($C_{\text{magnetite}}$) but not the sulfide (C_{sulfide}), especially given that the authors hope to calculate the volatile-sulfide but not volatile-magnetite partition coefficient.....and why the result of this division should be “+1” in this calculation. In addition, this equation can be changed to:

$$K_{(D \text{ magma volatile/sulfide})} = (C_{\text{Magnetite}} + C_{\text{Sulfide}} - C_{\text{Magnetite}}) / (C_{\text{Magnetite}}) = (C_{\text{Sulfide}}) / (C_{\text{Magnetite}})$$

I do not understand the theory of this equation, and why the ratio between C_{sulfide} and $C_{\text{magnetite}}$ equals to the volatile-sulfide partition coefficient although volatile is not mentioned in this equation? The calculation here is too crude to be easily understood by the readers, and I guess there is something wrong in this calculation.

Then, let me go deep into this calculation. The authors’ assumption that “magmatic volatile metal concentration is inferred as the difference between the sulfide and the replacing magnetite” sounds right if the whole system follows the mass balance, but their calculation ($C_{\text{volatile}} = C_{\text{sulfide}} - C_{\text{magnetite}}$, Ps. I have rebuilt their table, and they just did this simple subtraction) may be wrong. Here, the C_i is the weight percent of element i in phase, but not the mass of element i . If the masses of sulfide, magnetite and volatile do not hold the same value, the concentration of i in volatile can not equal to $C_{\text{sulfide}}^i - C_{\text{magnetite}}^i$ under the assumption of mass conservation. Let me take a simple example, if 30 g sulfide contains 100 ppm Cu, 30 g magnetite has 20 ppm Cu, and then the releasing mass of Cu into the volatile equals to $100 \times 30 - 20 \times 30 = 2400 \text{ g} \cdot \text{ppm}$. If the mass of volatile is about 10g, and then the Cu content should be $2400 / 10 = 240 \text{ ppm}$, but not the result of $C_{\text{sulfide}}^i - C_{\text{magnetite}}^i = 100 - 20 = 80 \text{ ppm}$ that is adopted in this manuscript. Because the density of volatile is extremely lower than that of sulfide, the mass of volatile must be far lower than that of sulfide/magnetite, and hence the metal concentration in volatile must be extremely underestimated if using the authors’ equation. On the other hand, the oxidation and breakdown of sulfide via a reaction of the form: $3 \text{ [FeS]}_{\text{Sul}} + 2 \text{H}_2 \text{O}_{\text{fluid}} + 2 \text{O}_{(2 \text{ fluid})} \rightarrow \text{[Fe]}_3 \text{O}_4 + 2 \text{H}_2 \text{S}_{\text{fluid}} + \text{[SO]}_{(2 \text{ fluid})}$, which is also cited by the authors in this manuscript. But the masses of 3FeS and Fe₃O₄ in both sides of this reaction are not the same, and hence we even can not assume that the sulfide and magnetite have the same mass during oxidation and breakdown, which further complicate the calculation of metal concentration in magmatic volatile. Although the authors have restated that “calculated values are considered as first order estimates due to their intrinsic uncertainties”, I do not think the misunderstanding of theory and fundamental errors in this calculation can be considered as “intrinsic uncertainties”, and even be tolerated and accepted for a publication in Nature Communications.

2. About the model of compound drop in magma mixing and mingling:

In line 59, “we develop a model in which compound drops from mafic enclaves coalesce and accumulate

into sulfide ovoids at the mafic-felsic melt interface". But after reading the whole manuscript, I think this model is broad-brush, and the readers may not get enough information about this model. Furthermore, this model introduces some confusions for me. In the classical model from Plail et al. (2018) which is also cited by authors here, an initially thin hybrid boundary layered forms at the interface of injecting basalt and dacitic magma at the bottom of magma chamber, and mafic enclaves will be formed when this hybrid boundary is sufficient buoyant to form plume (relates to the Rayleigh number). But the authors propose the hybrid boundary between small-scale mafic enclaves and dacite magma is the place for the coalescence of compound drop. This thin hybrid boundary around mafic enclave may be not stable during the dynamic magma mixing and mingling, especially given that dynamics of enclave may be highly variable, which occurs along with the continuous breakdown and refresh of melt in the thin hybrid boundary. This dynamic environment may be not suitable for the coalescence of compound drop. On the other hand, the authors states that the "strong viscosity and density contrast between andesitic and dacitic melts could have acted as a physical barrier, triggering compound drops coalescence". In my opinion, the density contrast between andesitic and dacitic magmas may be not too large, otherwise the hybrid boundary will easily obtain enough buoyant (high Rayleigh number) to induce the chaotic and turbulent convection in the whole magmatic system or just around the mafic enclaves, which is not suitable for the coalescence of compound drop. The following article may further the understanding of magma mixing and mingling, and may be helpful for the revision.

Spera, F.J., Schmidt, J.S., Bohron, W.A., Brown, G.A., 2016. Dynamics and thermodynamics of magma mixing: insights from a simple exploratory model....

3. About the destabilization of compound drops:

In the schematic figure 3, the authors suggest that the major volatile part will be divided into different small bubble, and then released into the magmatic system. But in the numerical models from Yao and Mungall (2020) and Rosenfeld et al. (2011), the destabilization of compound drop occurs with a very clear separation between sulfide and volatile droplets, which is distinct from the model mentioned here. Although the authors think that the variation of melt-volatile surface tension may drive the destabilization of compound drop, in my opinion, the surrounding fluid field and the dynamics processes around compound drop should be more important for the destabilization of compound drop. If the surrounding melt has a large flowing velocity, the compound drop may have a high capillary number, and then the dense sulfide can be easily separated from vapour phase. More information can be found in Rosenfeld et al. (2011) and Yao et al., (2019).

Rosenfeld, L., Lavrenteva, O.M., Spivak, R., Nir, A. 2011. Deformation of a partially engulfed compound drop slowly moving in an immiscible viscous fluid....

Yao, Z., Mungall, J., Qin, K., 2019. A preliminary model for the migration of sulfide droplets in a magmatic conduit and the significance of volatiles....

Minor comments:

Line 23: delete "fully"?

Line 24: change “calculation” by “first-order estimate”?

Line 63: should be “behaviors”

Line 65: delete “which”?

Line 87: “massive sulfide” or “massive ore”?

Line 120: change “than” by “the”

Line 150-151: the gradual boundary between andesite enclave and dacite host rock in Fig. 1b does not show the formation of a foamy intermediate hybrid layer. I can not observe the third layer in the Fig. 1b. I know this may be quite subjective, and you can ignore.

Line 154: Because you suggest that the temperature of dacitic melts is about 750-900 C, the corresponding melt viscosity in the figure 5.4 from Lesher and Spera, 2015 should be about 106-108 Pas. I must note that here these values are for the dynamic viscosity of silicate melts, but not the silicate magmas. The magma is a mixture of silicate melt, minerals, sulfide and volatile, and hence the dynamic viscosity of andesitic and dacitic magmas should be higher than the value mentioned by the authors here.

Line 207: “a highly efficient method” there should be a noun after the “appears thus as”?

Line 300: the name of journal is missed for this reference

Line 357: replace “A” by “the right part”

Line 368: “trace metal contents”

Reply to review for manuscript NCOMMS-22-45611-T

REVIEWER COMMENTS

Reviewer #1 (Remarks to the Author):

Manuscript NCOMMS-22-45611-T “Sulfur and chalcophile metal transfer via sulfide-volatile compound drops during magma mixing: evidence from the Christiana-Santorini-Kolumbo volcanic field” by C.G.C Patten, S. Hector, S.P. Kiliyas, M. Ulrich, A. Peillod, A. Beranoaguirre, P. Nomikou, E. Eiche, J. Kolb.

This is a very interesting paper dealing with the transfer of sulfur and chalcophile elements from the magma to the fluid phase, via the formation of sulfide-volatile compound drops. The manuscript illustrates possible natural examples of compound drops occurring in andesitic magmas mixing with dacitic magmas, which were observed in the eruptive products of the Christiana-Santorini-Kolumbo volcanic system. The manuscript is well written and clear, the figures are of high quality. Given the topic and the quality of the paper, I recommend it for publication in Nature Communications, after moderate revisions.

I suggest the authors to improve the discussion of two points in order to strengthen the paper: (i) the timing of sulfide oxidation and the mechanism operating it, and (ii) the calculation and the interpretation of volatile-sulfide partition coefficients.

The timing of sulfide oxidation and the mechanism operating it are not fully elucidated in the present version of the manuscript. It is unclear whether the fluids forming the compound drops are the same operating sulfide oxidation. Figure 3 shows a continuum between stages 2 and 3, but the occurrence of pyrite replacement along fractures would rather suggest the existence of two different events. Why the sulfides are oxidized? As the manuscript is short, in my opinion there is room for discussing these points, on the basis of textural observations. Several possibilities could be presented, for instance:

- sulfide oxidation could be due to a change in volatile composition due to magma mixing (volatiles from the dacitic magma would need to be more oxidizing, why?);
- sulfide transformation into magnetite could occur as a consequence of desulfurization during degassing (as described in Iacono-Marziano et al. 2022);
- sulfide oxidation could occur in a later stage (what could be the origin of these late fluids?).

Sulfide oxidation could be in indeed related to several processes but we think the main process for the reaction is the second option suggested by the reviewer, which is also the one described in the manuscript. Nadeau et al. (2010) have already well described the oxidation of pyrrhotite into magnetite or into pyrite and magnetite depending on the oxidation state:

The oxidation of pyrrhotite into pyrite and magnetite highlights that the oxidation process can occur as a single stage and a multi-stage model is not necessary. Nadeau et al. (2010) also associated the oxidation of the sulfides to magmatic degassing but due to hydrothermal fluid circulation. It should be pointed out that at the time of Nadeau et al. (2010), volatile-sulfide compound drops were not truly recognized as such.

Relative to the first possibility pointed out by the reviewer, we do not know if magma mixing led to a significant change in the volatile composition and triggered sulfide oxidation. Magmatic differentiation in arc systems generally leads to magnetite crystallization and to increasing reducing conditions (e.g. magnetite crisis; Jenner et al., 2010). We do observed magmatic magnetites in samples from Kameni and Kolumbo volcanos (see supplementary data) which should drive the redox state of the melts towards more reducing conditions. Hence, it appears unlikely that magma mixing with more evolved melt affects the redox of the volatiles towards more oxidizing conditions.

Relative to the third possibility, arc volcanos are dynamic systems where complex magmatic-hydrothermal processes occur. Indeed a later stage of hydrothermal fluids could have circulated through the rocks leading to sulfide oxidation. Such process, however, would have also affected the host rock and led to significant silicate alteration, possibly similar to porphyry deposits. Some of the sulfides observed in this study are fully replaced to magnetite while the host rock is fresh without signs of extensive hydrothermal alteration (e.g. Fig. 1E).

Altogether, we think that the oxidation process is related to one single stage due to “S degassing” happening within the compound drops, from the sulfide mineral to the volatile phase. Such process is similar to the model of Edmonds and Mathez (2017). The overall oxidizing conditions of arc melts (FMQ>1, Jugo et al., 2009) suggest that the volatile phases nucleating on the sulfides would allow for the oxidation of the latter. The difference in redox with ultramafic systems is likely a reason why the sulfides phases at Norilsk’, for example, are not oxidized (e.g. LeVaillant et al., 2017; Barnes et al., 2019).

I am also concerned by the direct comparison between volatile-sulfide partition coefficients calculated in this study and vapor-melt and brine-melt partition coefficients, as shown in Fig. 2. I do not think this comparison is meaningful, and I would like to see a more convincing evidence of the efficiency of compound drops in transferring chalcophile elements into shallow magmatic-hydrothermal arc systems. I also suggest to explain why volatile-sulfide partition coefficients are calculated with the formula at line 187, this is unclear.

By comparing vapor-sulfide partition coefficients with vapor-melt partition coefficients we can access the efficiency of metal mobility during sulfide oxidation relative to “classic” magma degassing. Similarly, Edmonds et al. (2018) compare emanation coefficients from various volcanoes with vapor-melt and sulfide-melt partition coefficients. We would like to the keep the comparison, but if the reviewer has some further arguments that with have missed for not comparing the partition coefficients, we will gladly reevaluate. The calculations of the partition coefficients are detailed extensively in reply to reviewer #3.

Minor comments:

Line 73: I would remove “Evidence ...is defined by”, and simply speak about two sulfide populations.

This section has been rewritten following reviewer 2 comments.

Line 75: Fig. 1a is cited as illustrating the micrometric sulfide blebs, I do not consider it as the most appropriate.

Corrected, now only Fig. 1c is called.

Line 112: Fig. 2b, instead of Fig. 2a?

Yes indeed, corrected.

Fig. 2: please correct KD sulfide-volatile in the figure legend

The figure 2 has been modified.

Reviewer #2 (Remarks to the Author):

Review of manuscript NCOMMS-22-45611-T

Title: Sulfur and chalcophile metal transfer via sulfide-volatile compound drops during magma mixing: evidence from the Christiana-Santorini-Kolumbo volcanic field

By C.G.C. Patten¹, S. Hector¹, S. P. Kiliyas², M. Ulrich³, A. Peillod¹, A. Beranoaguirre¹, P. Nomikou², E. Eiche^{1,4}, J. Kolb¹

Submitted to Nature Communications

In this manuscript, authors describe sulfide-oxide-vesicles associations observed within samples from the Christiana-Santorini-Kolumbo (CSK) volcanic field. These associations are interpreted as ‘fresh’ and oxidized ‘compound drops’. The authors interpret larger millimetric sulfide ovoids as being the result of coalescence of smaller compound drops, and the oxides (magnetite) as being the result of oxidation of primary magmatic sulfides during sulfide-volatile interaction. They then study metal compositions of the sulfides and the oxide phases in order to evaluate magmatic volatile-sulfide partition coefficients. They finally use all these observations to infer the potential impact of 1) sulfide droplet – volatile association, 2) the coalescence of these compound drops and 3) the oxidation of these sulfides on S and metal transfer mechanism between the sulfides and shallow magmatic-hydrothermal fluids in arc systems and therefore ore genetic processes in arc systems. Finding evidence of such processes in volcanic rocks can be challenging, especially being able to quantify and observe the actual transfer of metals from sulfide liquid to magmatic-hydrothermal fluids, and this would be one of few studies doing so. These fluids would then have the potential to become mineralizing fluids for hydrothermal Cu-

systems, which has significant implications on the genetic models for hydrothermal mineralized systems. However, I think the dataset presented here, in its actual form, does not provide strong enough evidence for such a claim. In order to evaluate whether it could, the authors would need to provide complementary information.

1. Quality of the data, Analytical approach, Data and methodology

The data presented seems to be of high quality with appropriate techniques and methods used. However, the authors need to present the data in its integrity. For example, sulfide composition data presented in supplementary material is for specific sulfides (pentlandite, pyrrhotite, chalcopyrite, etc...), and is only showing medians. To truly evaluate the quality of the manuscript we would need the whole dataset in supplementary material:

- all analyses collected as part of this study, not just median values, along with context (sulfide bleb vs ovoid, sulfide in inclusion?, etc....).

All single analysis are now presented in the additional supplementary table 5 with the type of mineral phase analyzed. Additional micro photographs of magmatic sulfide blebs associated with vesicles and variably oxidized sulfide ovoids are added to the supplementary figures 1 and 2.

- Error bars (uncertainties on all the analyses) need to be added to be able to do the comparisons they are doing.

Uncertainties on the analyses are also presented in the supplementary table 5. We do not show the uncertainties on Fig. 2 as it would become illegible. Showing the median, upper quartile and lower quartile in the graphs provide the necessary information on element concentrations in each mineral phase and allow for comparison.

- More details on the methods need to be added in supplementary material (referring to Nadeau et al 2010's supplementary material is too complicated for reviewers and future readers). Such as How average sulfide bleb compositions were calculated using individual phases' composition – see Supp. Table 2.

Just for clarity, only the Fe, used as internal standard for LA-ICP-MS, is recalculated. It is recalculated using equation (1) of the revised appendix which follows Nadeau et al. (2010) approach. This is detailed in the appendix.

- Better contextual information for the samples is needed, especially precise location (schematic section through the system showing where the samples come from, map?): How many samples were studied as part of this work exactly? How representative of the overall system are they? Are these two 'populations' only described on the basis of 2-3 samples that have been described in detail here? If so, then this does not seem statistically representative enough. Are there more observations of these two sulfide populations in the system as a whole, within the samples, within the dataset?

- Overall, a detailed sample description and location information needs to be added. We currently cannot evaluate how representative the selected samples are of the system and whether or not the amount of observation and data collected is enough to be significant.

Preservation of fresh sulfide-volatile compounds in rocks is scarce and, as stated by the reviewer, finding these rocks is challenging. During the sampling campaign of this study we were not specifically targeting sulfide-volatile compounds but rather mafic enclaves. Their finding happened by serendipity and at this stage we cannot determine how representative they are of the system. Further fieldwork is needed. From the Nea Kameni volcano we found 4 samples with petrographic evidence related to compound drops (magmatic sulfide blebs, variably oxidized sulfide ovoids) out of 5 samples with mafic enclaves and from the Kolumbo volcano we found 2 sample with compound drops out of 11 with enclaves; it should be emphasized that these are ROV samples which were recovered neither for sulfide-volatile compounds or enclaves. The discovery by serendipity of sulfide-volatile compounds in 6 samples out of 16 suggests that they are relatively common.

1. Level of support for the conclusions

There are many interpretations and conclusions present in this manuscript that need stronger support:

- Compound drops

The fact that what we are observing are indeed compound drops needs to be better proven (rather than stated). This manuscript needs to add a section demonstrating the fact that what they are describing as ‘compound drops’ are indeed preserved/frozen in place sulfide bleb - gas bubble pairs. Not just refer to supplementary material (that does not really demonstrate it either in the text). Not simply compare the sulfide blebs with sulfide blebs from other described compound drops in the literature, but actually describe and compare with literature compare the whole sulfide-gas bubble association/pair.

Following the reviewer recommendation, this section has been reworked with emphasis on the sulfide-volatile pair description. For clarity the supplementary discussion stated in the text is for the sample whole rock geochemistry and is not related to the compounds per se. The text has been modified to avoid confusion.

- Sulfide composition:

Not enough data to support the compositional comparison between the sulfides from this study with MORB sulfides and compound drops sulfides (See figure 2A). More data (from literature) needs to be added to support the comparison and conclusion.

As for the data collected as part of this study, we need more information as well: 1) How many sulfide bleb inclusions in plagioclase were analysed? 2) composition shown in the diagram is this the ‘evaluated’ composition of the whole bleb? How was that determined? Or is it the composition of specific sulfide phases within the bleb?

When comparing composition of sulfides in sulfide bleb with sulfides in ovoid, you need to compare the same thing, i.e. composition of individual phase type, or compare the calculate composition of the overall sulfides. This is currently not the case in figure 2.

We have added more literature data to Figure 2a (Peach et al. 1990; Patten et al. 2013; Keith et al. 2017; Zelenski et al. 2017; Fulignati et al. 2018). Although EMPA major element data for small sulfide blebs in different settings are relatively common, in-situ trace element analyses are

scarce due to their relatively small size which impair in-situ LA-ICP-MS analysis. The data presented in Fig. 2a are from four sulfide blebs which were large enough for LA-ICP-MS. Differentiated sulfide bleb compositions are not perfectly matching those of primary magmatic sulfide liquids (Patten et al., 2012), but they are assumed to be representative of the whole primary magmatic sulfide liquid and do not represent individual sulfide phases (e.g. only Mss or Iss).

For the partition coefficient calculations we have calculated the bulk magmatic sulfide compositions assuming different proportions of Po, Cpy and Mag (See detail below with comments from reviewer 3) which can be indeed compared with magmatic blebs composition (See figure below). The calculated magmatic compositions and the magmatic sulfide blebs have very similar element concentration profiles despite some concentration differences for a few elements (U, In, Th and Sb); highlighting the genetic link between the magmatic blebs and the sulfide ovoids. This comparison cannot be made in Fig. 2a as there would be too much information. It is however added to the supplementary discussion.

Figure 1. Comparison of trace metal composition of magmatic sulfide blebs and recalculated magmatic sulfide liquid.

- Evidence of oxidation process of the sulfides?

Can the authors use mineral maps and mineral compositions to do a rough mass balance for the oxidation process of the sulfides?

Due to density differences between the pyrrhotite (dominating sulfide phase, $d=4.61 \text{ g.cm}^{-3}$) and the magnetite ($d=5.15 \text{ g.cm}^{-3}$) a decrease in volume occurs during oxidation of pyrrhotite to magnetite. This is shown by the relatively high porosity in the magnetite (Fig. 1d, g; supplementary figure 2f). Mass balance could be calculated but we think that the outcomes on element mobility will be the same as that from the calculation of the partition coefficients and are therefore not necessary.

2. Potential significance of the results

Many of the processes and implications described in this paper have already been stated in previous papers (especially in Nadeau et al 2010, Mungall et al 2015, Edmonds and Mathers 2017, Edmonds et al 2018) and aren't novel as such. However, actual evidence in the rocks of the processes described here (presence of compound drops in andesite enclaves, coalescence of sulfide-bubble compound and especially metal transfer into magmatic hydrothermal fluids through the oxidation process of the sulfides) would be very significant.

In conclusion, this is a manuscript that reads well and describe a very interesting process and its potential implications. The approach is valid and the data is generally well presented. However, in its current form, interpretations and conclusions are not well enough supported by the data (or at least not in the state in which it is provided).

I have also added numerous reviews, comments, and suggestions to improve this manuscript on the attached annotated versions of the files provided for the review.

Additional comments by reviewer 2 from supplementary materials:

What are these? gas bubbles? within the interpreted vesicle? Which sample does this photomicrograph come from?

These are air bubbles within the epoxy of the thin sections which is filling the vesicle hosting the remnants of compound drop. This picture comes from sample TH-20-23 as noted on the bottom left of the microphotograph. In this sample the ovoids are fully oxidized with only minor pyrite preserved within the magnetite.

Add error bars

We rather not as it would make the graph illegible, especially with the log scale. We think that the shaded areas representing the upper and lower quartile are enough to show the trace element concentration variations.

What does this refer to exactly? magmatic magnetite from this study here? where from?

Yes these are magmatic magnetite from the Kameni and Kolumbo volcanoes which are present in the mafic enclaves and which are not related to the ovoids. These are highlighted in blue in supplementary figure 3C. These magnetites have subeuhedral texture and have high Ti, Ni, V and Co concentrations which is specific of magmatic magnetite.

still same sample? TH-20-30A ?

Yes, this is added.

error in sample name? shouldn't it be TH-20-30-2 ?

Yes indeed, thanks. The name has been changed.

sample name?

The sample name has been changed.

Reviewer #3 (Remarks to the Author):

This is a relatively concise and potentially meaningful contribution to the discussion on the vertical transport of sulfur and metals in the magmatic plumbing system. The formation and dynamics of sulfide-vapour compound drop become one of the hot research topics in recent years since the groundbreaking work of Mungall, and here the authors provide high-resolution X-ray fluorescence mapping and in-situ mineral analysis of compound drops found in the Christiana-Santorini-Kolumbo volcanic field. The estimation of the magmatic volatile-sulfide partition coefficients of metal elements based on their concentrations within sulfide and secondary magnetite is a novel idea, but I must note that the calculation behind this idea seems to be too crude, or even not right if just reading their manuscript and supplementary information. Secondly, I guess this manuscript was transferred from other journal, and thus there are not so much information in the short manuscript, which weakens its reliability. The geological setting of CSK volcanic field was introduced in the supplementary information, but it just occupies half of the page and can be moved to the major manuscript. In addition, all information about the magma evolution and mixing of CSK volcanic field is a summary of previous articles, and no new idea, data and information is reported in the supplementary information. Hence, this manuscript is a little thin on content, especially given that their new idea about the magmatic volatile-sulfide partition coefficients is based on an overly simplistic calculation. Thirdly, the dynamics of compound drop during magma mixing and mingling should be the focus of this work, but the associated discussion and interpretation are likely qualitative conjecture, which may not deep our current understanding of this topic. Overall, this is an interesting and ordinary article about the sulfide-vapour compound drop, but it may have a long way to reach the high level of Nature Communications. My recommendation as of this reading is, narrowly, major

revision, but would understand if the suggested revisions cannot be done in the time required for a revision and thus a rejection with resubmission is required, and will leave that to the judgment of the editor.

We are thankful for the constructive reviews. A section about the geological setting is now added to the manuscript. Regarding the general magma evolution and mixing of the Kolumbo volcano, we think that bringing new idea to this topic is beyond the aim of this study. The provided information relative to magma evolution and magma mingling are here for the readers to understand the framework within which compound drops form and evolve. The partition coefficients and the dynamics of compound drops during mixing are discussed more in details below.

Major comments:

1. Model for the calculation of magmatic volatile-sulfide partition coefficients:

In the manuscript, “the magmatic volatile-sulfide partition coefficients are calculated as follows”:

$$K_{(D \text{ magma volatile/sulfide})} = 1 + (C_{\text{Sulfide}} - C_{\text{Magnetite}}) / (C_{\text{Magnetite}})$$

But the authors had not provided any theoretical explanation about this equation. They may assume that the metal concentration in magmatic volatile is equals to the difference between the sulfide and the replacing magnetite (i.e., $C_{\text{volatile}} = C_{\text{sulfide}} - C_{\text{magnetite}}$), but why this value should be divided by the metal concentration of magnetite ($C_{\text{magnetite}}$) but not the sulfide (C_{sulfide}), especially given that the authors hope to calculate the volatile-sulfide but not volatile-magnetite partition coefficient.....and why the result of this division should be “+1” in this calculation. In addition, this equation can be changed to:

$$K_{(D \text{ magma volatile/sulfide})} = (C_{\text{Magnetite}} + C_{\text{Sulfide}} - C_{\text{Magnetite}}) / (C_{\text{Magnetite}}) = (C_{\text{Sulfide}}) / (C_{\text{Magnetite}})$$

I do not understand the theory of this equation, and why the ratio between C_{sulfide} and $C_{\text{magnetite}}$ equals to the volatile-sulfide partition coefficient although volatile is not mentioned in this equation? The calculation here is too crude to be easily understood by the readers, and I guess there is something wrong in this calculation.

Then, let me go deep into this calculation. The authors’ assumption that “magmatic volatile metal concentration is inferred as the difference between the sulfide and the replacing magnetite” sounds right if the whole system follows the mass balance, but their calculation ($C_{\text{volatile}} = C_{\text{sulfide}} - C_{\text{magnetite}}$, Ps. I have rebuilt their table, and they just did this simple subtraction) may be wrong. Here, the C_i is the weight percent of element i in phase, but not the mass of element i . If the masses of sulfide, magnetite and volatile do not hold the same value, the concentration of i in volatile can not equal to $C_{\text{sulfide}}^i - C_{\text{magnetite}}^i$ under the assumption of mass conservation. Let me take a simple example, if 30 g sulfide contains 100 ppm Cu, 30 g magnetite has 20 ppm Cu, and then the releasing mass of Cu into the volatile equals to $100 \times 30 - 20 \times 30 = 2400$ g•ppm. If the mass of volatile is about 10g, and then the Cu content should be

2400/10= 240 ppm, but not the result of $C_{\text{sulfide}} - C_{\text{magnetite}} = 100 - 20 = 80$ ppm that is adopted in this manuscript. Because the density of volatile is extremely lower than that of sulfide, the mass of volatile must be far lower than that of sulfide/magnetite, and hence the metal concentration in volatile must be extremely underestimated if using the authors' equation. On the other hand, the oxidation and breakdown of sulfide via a reaction of the form: $3[\text{FeS}]_{\text{Sul}} + 2\text{H}_2\text{O}_{\text{fluid}} + 2\text{O}_{(2\text{ fluid})} \rightarrow [\text{Fe}]_3\text{O}_4 + 2\text{H}_2\text{S}_{\text{fluid}} + [\text{SO}]_{(2\text{ fluid})}$, which is also cited by the authors in this manuscript. But the masses of 3FeS and Fe₃O₄ in both sides of this reaction are not the same, and hence we even can not assume that the sulfide and magnetite have the same mass during oxidation and breakdown, which further complicate the calculation of metal concentration in magmatic volatile. Although the authors have restated that "calculated values are considered as first order estimates due to their intrinsic uncertainties", I do not think the misunderstanding of theory and fundamental errors in this calculation can be considered as "intrinsic uncertainties", and even be tolerated and accepted for a publication in Nature Communications.

We agree that the equation presented in the manuscript is too crude and needs more explanations; we are also thankful for the detailed comments which allow to improve these calculations. Calculation of the volatile-sulfide partition coefficients is challenging because, 1) the volatile phase is not preserved, 2) the sulfides are not at equilibrium with the volatile phase and are oxidized to magnetite and 3) the compound drop system is not a closed system and is bound to interact with the surrounding silicate melt. The compound drop system is considered at constant temperature and pressure with two main stages: 1) an initial stage with the sulfide and the volatile phases and which are not at equilibrium and, 2) a final stage with magnetite and the volatile phase at equilibrium after full oxidation of the sulfide (see figure below). Over whole mass conservation of the system is assumed between the two stages. We also assume that the initial volatile phase in contact with the sulfide phase is free of the elements of interest. Finally the system is not considered closed with sulfide-melt, magnetite-melt and volatile-melt interactions in addition to sulfide-volatile interaction.

Sulfide-melt interaction is assumed to have occurred early during the initial stages of magmatic differentiation when the sulfur content at sulfide saturation is reached. At the time of compound drops accumulation and coalescence into ovoids, it is assumed that the sulfide liquid is close to equilibrium with the silicate melt. Volatile-melt interaction, on the other hand, is still likely to occur while sulfides are oxidized to magnetite. Hence the final concentration of the volatile phase is actually dependent on sulfide-volatile and melt-volatile interactions. The volatile-melt interaction is, however, not an issue for the calculations of emanation and partition coefficients as we are using the sulfide and oxide concentrations; if the volatile phase was preserved and its element concentrations known, such interaction would have been problematic in calculating the partition coefficients.

Figure 2. Schematic of the sulfide-volatile compound system for calculation of partition coefficients. X_{sulf}^i , X_{magn}^i , $X_{vol-init}^i$ and $X_{vol-fin}^i$ are the mole fractions of the element (i) for the sulfide, magnetite, initial volatile and final volatile phases, respectively.

For understanding metal mobility and calculating the partition coefficient we now use a two steps approach, first we determine the “emanation coefficient”, using the method from Edmonds et al. (2018) and Lambert et al. (1985), and then we calculate partition coefficients.

The emanation coefficients (Edmonds et al., 2018; Lambert et al., 1985), although applied to melts and post-eruptive lavas, are particularly well suited here as it uses only the initial concentration of the system (in our case the sulfides) and the final concentration of the system (in our case the magnetite). The emanation coefficients for the compound drops, expressed in percent, can be defined as:

$$Ec_i = \left(\frac{C_{i\ sulf}}{d_{sulf}} - \frac{C_{i\ mag}}{d_{mag}} \right) / \left(\frac{C_{i\ sulf}}{d_{sulf}} \right) * 100 \quad (1)$$

with $C_{i\ sulf}$ and $C_{i\ mag}$ the concentration of a given element in the magmatic sulfides and in the replacing magnetite, respectively and d_{sulf} and d_{mag} the density of the magmatic sulfides and replacing magnetite respectively. The concentration and the density of the magmatic sulfides are calculated as:

$$C_{i\ sulf} = C_{i\ Po} * X_{Po} + C_{i\ Cpy} * X_{Cpy} + C_{i\ Mag} * X_{Mag} \quad (2)$$

$$d_{sulf} = d_{Po} * X_{Po} + d_{Cpy} * X_{Cpy} + d_{Mag} * X_{Mag} \quad (3)$$

with $C_{i\ Po}$, $C_{i\ Cpy}$ and $C_{i\ Mag}$ the concentration of the element i in the pyrrhotite, chalcopyrite and magnetite which have crystallized from the initial magmatic sulfide liquid, X_{Po} , X_{Cpy} and X_{Mag} the fraction of pyrrhotite, chalcopyrite and magnetite which have crystallized from the magmatic sulfide liquid and d_{Po} , d_{Cpy} and d_{Mag} the density of pyrrhotite, chalcopyrite and magnetite. The pyrrhotite and chalcopyrite concentrations are from the analysis of ovoids (this study) while the magnetite concentrations are from Dare et al. (2012). Magnetite crystallization from a sulfide

liquid is common (Czamanske and Moore, 1977; Peach et al., 1990; Patten et al., 2012; Dare et al., 2012) and needs to be taken in account for the calculations. The use of magnetite concentrations from Dare et al. (2012) as a proxy for our system, however, has limitations because of the radically different nature of the magmatic environment (i.e. Sudbury vs CSK volcanic field). This study, however, is the only one known to the authors for having measured by LA-ICP-MS magnetites which have crystallized from a magmatic sulfide liquid. $C_{i\ sulf}$ and $d_{i\ sulf}$ are calculated assuming various proportion of pyrrhotite and chalcopyrite in the initial magmatic sulfide phase (X_{Po} : 0.85-0.65, X_{Cpy} : 0.1-0.3) with a fixed fraction of magnetite (X_{Mag} : 0.05).

The emanation coefficients are presented in the supplementary table 4 and are also plotted on Fig. 2c. Importantly, the emanation coefficients can have negative values, because the compound-drop system is not closed and isolated from the silicate melt and elements, especially those compatible in magnetite, can partition into the magnetite during sulfide oxidation.

We now calculate the volatile-sulfide partition coefficient. As a first approach, one would define the volatile-sulfide partition coefficient as:

$$K_{D\ volatile/sulfide}^i = \frac{C_{volatile}^i}{C_{Sulfide}^i} \quad (4)$$

As described in the manuscript, however, the sulfides are not in equilibrium with the volatile phase, but rather with the magnetite as oxidation occurs. Hence, at equilibrium, the partition coefficient should be written as:

$$K_{D\ volatile/sulfide}^i = \frac{C_{volatile}^i}{C_{magnetite}^i} \quad (5)$$

In this equation the $C_{volatile}^i$ is unknown, and, as point out by the reviewer, the assumption that it is the difference between the sulfide and oxide concentration ($C_{Sulfide}^i - C_{magnetite}^i$) does not stand because it does not consider mass balance. However, partition coefficients can also be calculated using mole fractions (which are mass balanced):

$$K_{D\ volatile/sulfide}^i = \frac{X_{volatile}^i}{X_{magnetite}^i} \quad (6)$$

With $X_{volatile}^i$ the mole fraction of the volatile phase and $X_{magnetite}^i$ of the magnetite (see figure). Also the system can be described in the initial stage as:

$$X_{sulfide}^i + X_{volatile\ initial}^i = 1 \quad (7)$$

The initial volatile phase is considered to be free of the elements of interest and is therefore null. Equation (4) is simplified to:

$$X_{sulfide}^i = 1 \quad (8)$$

In the final stage, the system can be described as:

$$X_{magnetite}^i + X_{volatile\ final}^i = 1 \quad (9)$$

Therefore the relationship between the initial stage and the final stage can be written as:

$$X_{sulfide}^i = X_{magnetite}^i + X_{volatile\ final}^i \quad (10)$$

Meaning that for a given element (i) the total mole fraction of the system is initially controlled by the sulfide phase and it is then partitioned between the magnetite and the final volatile phase as oxidation occurs. The volatile-melt interaction here does not need to be considered. Additionally, it is assumed that at equilibrium both the magnetite and the volatile phases cannot be devoid of any given element, implying $0 > X_{magnetite}^i < 1$ and $0 > X_{volatile\ final}^i < 1$.

Because the system is described with mole fractions, now the following relationship is true:

$$X_{volatile\ final}^i = X_{sulfide}^i - X_{magnetite}^i \quad (11)$$

And the partition coefficient can be written as:

$$K_D^{i\ volatile/sulfide} = \frac{X_{sulfide}^i - X_{magnetite}^i}{X_{magnetite}^i} \quad (12)$$

As $X_{magnetite}^i < X_{sulfide}^i$ the condition of $K_D^{i\ volatile/sulfide} > 0$ is preserved.

To calculate the partition coefficients, the mole fractions need to be converted to concentrations:

$$X_{Sulfide}^i = C_{Sulfide}^i * \frac{M_{total}}{M_{sulfide}} \quad (13)$$

$$X_{magnetite}^i = C_{magnetite}^i * \frac{M_{total}}{M_{magnetite}} \quad (14)$$

With $M_{sulfide}$ the sulfide molar mass, $M_{magnetite}$ the magnetite molar mass and M_{total} the average molar mass. The average molar mass can be defined from both the initial and final states of the system:

$$M_{total} = \sum_i X_{i_{initial}} M_i = X_{sulfide}^i * M_{sulfide} + X_{volatile\ intiale}^i * M_{volatile} \quad (15)$$

$$M_{total} = \sum_i X_{i_{final}} M_i = X_{magnetite}^i * M_{magnetite} + X_{volatile\ final}^i * M_{volatile} \quad (16)$$

The partition coefficients can then be develop:

$$K_D^{i\ volatile/sulfide} = \frac{X_{sulfide}^i - X_{magnetite}^i}{X_{magnetite}^i} \quad (17)$$

$$K_D^{i\ volatile/sulfide} = \left(C_{Sulfide}^i * \frac{M_{total}}{M_{sulfide}} - C_{magnetite}^i * \frac{M_{total}}{M_{magnetite}} \right) / \left(C_{magnetite}^i * \frac{M_{total}}{M_{magnetite}} \right) \quad (18)$$

$$K_D^i \text{ volatile/sulfide} = M_{total} * \left(\frac{C_{Sulfide}^i}{M_{sulfide}} - \frac{C_{magnetite}^i}{M_{magnetite}} \right) / M_{total} * \left(\frac{C_{magnetite}^i}{M_{magnetite}} \right) \quad (19)$$

$$K_D^i \text{ volatile/sulfide} = \left(\frac{C_{Sulfide}^i}{M_{sulfide}} - \frac{C_{magnetite}^i}{M_{magnetite}} \right) / \left(\frac{C_{magnetite}^i}{M_{magnetite}} \right) \quad (20)$$

The determined equation for the partition coefficients is akin to the “relative extraction efficiency” from Guo and Audétat (2017) which can be considered as a proxy for partition coefficients. Furthermore, the equation is also similar to the equation for the emanation coefficients but the denominator is the final state of the system rather than the initial state (and molar masses are used instead of densities). From this difference, when $C_{isulf} \gg C_{imag}$ the emanation coefficient tend towards 100% (full mobility in the volatile phase) while the partition coefficient has no limits and can reach high values (Fig. 2c), as expected for partition coefficients. As for the emanation coefficients, the partition coefficients can have negative values if $\frac{C_{magnetite}^i}{M_{magnetite}} > \frac{C_{Sulfide}^i}{M_{sulfide}}$, which should not occur if the system was closed. As discussed previously for the emanation coefficients, the compound drop system is not closed and is bound to interact with the silicate melt, with element diffusion from the silicate melt into the magnetite during sulfide oxidation. Finally, for clarity, in the previous draft, the system was considered closed and the equation was modified to:

$$K_D^i \text{ volatile/sulfide} = 1 + \left(\frac{C_{Sulfide}^i}{M_{sulfide}} - \frac{C_{magnetite}^i}{M_{magnetite}} \right) / \left(\frac{C_{magnetite}^i}{M_{magnetite}} \right) \quad (21)$$

Which prevented to have negative K_D . We think, however, that considering the system open is closer to the truth and thus, equation (20) is used for the calculations. The details about the emanation and partition coefficients are now added in the manuscript and in the supplementary discussion.

2. About the model of compound drop in magma mixing and mingling:

In line 59, “we develop a model in which compound drops from mafic enclaves coalesce and accumulate into sulfide ovoids at the mafic-felsic melt interface”. But after reading the whole manuscript, I think this model is broad-brush, and the readers may not get enough information about this model. Furthermore, this model introduces some confusions for me. In the classical model from Plail et al. (2018) which is also cited by authors here, an initially thin hybrid boundary layered forms at the interface of injecting basalt and dacitic magma at the bottom of magma chamber, and mafic enclaves will be formed when this hybrid boundary is sufficient buoyant to form plume (relates to the Rayleigh number). But the authors propose the hybrid boundary between small-scale mafic enclaves and dacite magma is the place for the coalescence of compound drop. This thin hybrid boundary around mafic enclave may be not stable during the dynamic magma mixing and mingling, especially given that dynamics of enclave may be highly variable, which occurs along with the continuous breakdown and refresh of melt in the thin hybrid boundary. This dynamic environment may be not suitable for the coalescence of compound drop.

We agree with the reviewer that thin hybrid layers forming solely at the andesite enclave margins would not be stable enough for the coalescence of the compound drops. Hence, are the andesitic enclaves observed in this study the remnants of a thick hybrid layer between the underplating mafic magma and the felsic magma at the base of the magmatic chamber (as suggested by the

model of Plail et al., 2018)? Interestingly the enclaves locally contain quartz veins which can help us answering the question. These quartz veins, as seen in figure 1e, are made up almost exclusively of quartz grains. They have grain size from few tens of microns up to hundreds of microns and contains fluids inclusions as well as discrete opaque minerals. The contact with the andesite host rock is relatively sharp with local host rock clasts assimilated within the vein. The nature of the veins appear to be hydrothermal. Importantly the quartz veins are cross-cut at the andesite-dacite interface by the dacitic rocks (Fig. 1e) and have not been observed within the dacitic rocks. This implies that veins are not related to late hydrothermal processes but are related to the magmatic-hydrothermal processes which occurred within the andesitic magma. Furthermore, magmatic magnetites are locally aligned along the veins within the andesitic enclaves, suggesting a sub-solidus process. The conditions of formation of these veins are cryptic and beyond the scope of this paper, but it appears unlikely that they form from small andesitic enclaves (up to several of centimeters) during magma mingling. Alternatively, if we consider the enclaves as fragments of the mafic-felsic magma interface at the bottom of the magmatic chamber, these veins could be interpreted as Si-rich residual fluids which have exsolved from the large mafic magma during cooling. Such process is somehow similar to the late residual silicate melt which partially fill the volatile fraction of the compound drops observed in the Norilsk' ultramafic system (Barnes et al., 2019). Interestingly in figure 1e, the quartz veins possibly fill some porosity related to the sulfide/oxide ovoids. Finally the relationship between the quartz veins and the compounds remnants in figure 1e does not imply that the quartz veins are oxidizing the sulfides as oxidized ovoids are observed in quartz vein-free samples. Locally to ovoids appear to be crosscut by the quartz veins (Fig. 1e) implying the quartz vein postdate the formation of the ovoids.

This is now detailed in the manuscript and figure 3 has been modified.

On the other hand, the authors states that the “strong viscosity and density contrast between andesitic and dacitic melts could have acted as a physical barrier, triggering compound drops coalescence”. In my opinion, the density contrast between andesitic and dacitic magmas may be not too large, otherwise the hybrid boundary will easily obtain enough buoyant (high Rayleigh number) to induce the chaotic and turbulent convection in the whole magmatic system or just around the mafic enclaves, which is not suitable for the coalescence of compound drop. The following article may further the understanding of magma mixing and mingling, and may be helpful for the revision.

Spera, F.J., Schmidt, J.S., Bohron, W.A., Brown, G.A., 2016. Dynamics and thermodynamics of magma mixing: insights from a simple exploratory model....

The use of “strong viscosity and density contrast between andesitic and dacitic melts” can be misleading as highlighted by the reviewer. The interface of mafic and felsic magmas is nevertheless a loci of volatile bubble accumulation under the right conditions, eventually leading the formation to a foamy hybrid layer (Plail et al., 2018; Edmonds and Woods, 2018). Additionally, magnetite-volatile aggregates, which could be analogues to sulfide-volatile compounds, have been found to also accumulate at the boundary between a basaltic and andesitic magmas (Edmonds et al., 2015). Hence, similarly to volatile bubbles and magnetite-volatile aggregates, sulfide-volatile compound drops will rise within the andesitic melt and stagnate at the melt interface. The argument has been changed in the text.

3. About the destabilization of compound drops:

In the schematic figure 3, the authors suggest that the major volatile part will be divided into different small bubble, and then released into the magmatic system. But in the numerical models from Yao and Mungall (2020) and Rosenfeld et al. (2011), the destabilization of compound drop occurs with a very clear separation between sulfide and volatile droplets, which is distinct from the model mentioned here. Although the authors think that the variation of melt-volatile surface tension may drive the destabilization of compound drop, in my opinion, the surrounding fluid field and the dynamics processes around compound drop should be more important for the destabilization of compound drop. If the surrounding melt has a large flowing velocity, the compound drop may have a high capillary number, and then the dense sulfide can be easily separated from vapour phase. More information can be found in Rosenfeld et al. (2011) and Yao et al., (2019).

Rosenfeld, L., Lavrenteva, O.M., Spivak, R., Nir, A. 2011. Deformation of a partially engulfed compound drop slowly moving in an immiscible viscous fluid....

Yao, Z., Mungall, J., Qin, K., 2019. A preliminary model for the migration of sulfide droplets in a magmatic conduit and the significance of volatiles....

We are thankful for the insight and the references. The model in Fig. 3 has been modified accordingly. To pinpoint the mechanism responsible for the compound drop separation, however, is beyond the scope of the study in our opinion. Nevertheless, the manuscript has been now modified following the suggestions.

Minor comments:

Line 23: delete “fully”?

Done

Line 24: change “calculation” by “first-order estimate”?

Done

Line 63: should be “behaviors”

This section has been rewritten.

Line 65: delete “which”?

This section has been rewritten.

Line 87: “massive sulfide” or “massive ore”?

The use of massive here is misleading as we cannot talk about massive sulfide or massive ore. It has been changed to “large grains”.

Line 120: change “than” by “the”

Done

Line 150-151: the gradual boundary between andesite enclave and dacite host rock in Fig. 1b does not show the formation of a foamy intermediate hybrid layer. I can not observe the third layer in the Fig. 1b. I know this may be quite subjective, and you can ignore.

The boundary texture in fig. 1b is indeed not really foamy (by opposition of the porous dacite in Fig. 1e) but can nevertheless be interpreted as a hybrid layer. Indeed the microXRF mineral maps allow to clearly distinguish between the dacitic matrix (red in Fig. 1b; high K in supplementary maps) and the andesitic matrix (green in Fig. 1b; low K in supplementary maps). The contact between the two matrices is not a sharp boundary clearly separating a dacite from an andesite. On the contrary the andesitic enclave (defined from petrographic observations based on porosity, phenocryst proportion and color) show patches with dacitic composition and thus showing a hybrid composition. The whole thin section, apart for the delimited porous dacite in the bottom left, might be actually part of the hybrid layer. The term of “foamy” has been removed in the manuscript to avoid confusion.

Line 154: Because you suggest that the temperature of dacitic melts is about 750-900 C, the corresponding melt viscosity in the figure 5.4 from Leshner and Spera, 2015 should be about 106-108 Pas. I must note that here these values are for the dynamic viscosity of silicate melts, but not the silicate magmas. The magma is a mixture of silicate melt, minerals, sulfide and volatile, and hence the dynamic viscosity of andesitic and dacitic magmas should be higher than the value mentioned by the authors here.

Following the reviewer’s previous comments, the paragraph with the argument about the density has been modified.

Line 207: “a highly efficient method” there should be a noun after the “appears thus as”?

Done

Line 300: the name of journal is missed for this reference

Done

Line 357: replace “A” by “the right part”

Done

Line 368: “trace metal contents”

Done

REVIEWER COMMENTS

Reviewer #1 (Remarks to the Author):

Manuscript NCOMMS-22-45611-A "Transfer of sulfur and chalcophile metals via sulfide-volatile compound drops in the Christiana-Santorini-Kolumbo volcanic field" by C.G.C Patten, S. Hector, S.P. Kilias, M. Ulrich, A. Peillod, A. Beranoaguirre, P. Nomikou, E. Eiche, J. Kolb.

In my previous review I suggested the authors to improve the discussion of two points in order to strengthen the paper:

- The timing of sulfide oxidation and the mechanism operating.

In their reply the authors present a long and comprehensive answer to my comment. I would have preferred that at least part of this answer was incorporated into the main text, it could be useful for other readers.

- The calculation and the interpretation of volatile-sulfide partition coefficients.

The text is much clearer now, but I suggest to be even more explicit in explaining (i) what the emanation and partition coefficients indicate (e.g. the order of volatility/mobilization of the investigated elements), and (ii) what the comparison with vapor-melt partition coefficients from the literature indicates (e.g. some elements are more efficiently mobilized by the oxidation of magmatic sulfides than by "normal" magmatic degassing). I would also clarify that the comparison is only qualitative (i.e. relative variations are only discussed, and not calculated E_c and K_d values).

Reviewer #2 (Remarks to the Author):

Review of revised manuscript NCOMMS-22-45611A

The authors address my concerns quite well in their responses. However, a few things still need to be considered:

- In the data provided for individual analysis (chemistry of individual phases - po, ccp, py, mag, and magmatic blebs) there is still in the table a lot of analyses with concentrations in certain elements that are well below the stated LOD for this specific analysis. shouldn't these simply be removed/edited? In particular (but not limited to), this would be important for the following elements and phases as they have an impact on the comparisons and interpretations made in the text): Au in Py and Po and Ti in ovoid magnetite.

- Some information given in the response to the reviews need to actually be added in the manuscript itself (in legends - text). Such as

- information on the number of analysed sulfide bleb inclusions, this should be added in the legend of Fig 2.
- the fact that magnetite compositions presented in Figure 2b are 'magmatic magnetite interpreted as crystallising from the silicate melt

Also in figure 2b and results presentation, the authors discuss Ni, Co and Mo enrichment of Py and Po compared to ccp. These is really not evident from the data presented in the figure itself.

Apart from the above comments, my main concerns about the manuscript have been answered by the responses to the reviews and edits to the manuscript. Even though I still feel like the story here would be a lot stronger if there was a more statistically representative number of samples/analyses (which does not seem to be possible to increase at this stage without going back on the field and sampling more)

Reviewer #3 (Remarks to the Author):

This manuscript introduces an interesting sulfide-oxide-vesicles associations occurring in andesitic magmas mixing with dacitic magmas, which was observed in the eruptive products of the Christiana-Santorini-Kolumbo volcanic system. The formation and coalescence of these compound drops can enhance the transfer of sulfur and chalcophile elements in arc magmatic-hydrothermal systems. Meanwhile, the sulfide will be oxidated to magnetite during this process, and the authors also estimate the partition coefficients of chalcophile elements between magnetite and volatile phase. Relative to the original manuscript, some instructive modifications had been made in the new revision under the previous comments and suggestions from the previous reviewers. However, I still think that some important concerns and confusions from the reviewers have not completely/clearly solved in this new revision. In my first review, I pointed out that the calculations of partition coefficients are not right, but the equations in the new revision are still vague or even wrong in the physical meaning, which will be explained in the following major comments. Secondly, the number of sulfide-oxide-vesicles associations observed in the CSK volcanic system is highly limited, and the interpretation and claim here, in my opinion, seem to step into an excessive amount of over explaining situation. Based on the current state of this revision, and the limited improvement after the first round of review, my recommendation as of this reading is, rejection.

Major comments:

The oxidation process of sulfide is still ambiguous for the readers.

As the point suggested by the reviewer 1, the mechanism of sulfide oxidation is not fully elucidated in the previous version of the manuscript. The authors have made an interpretation for this process within the "Reply to review for manuscript....", but the in the main part of the new revision, the associated interpretation is still limited and vague. The authors may have considered that they have cite the associated articles in the manuscript, but the sulfide oxidation process does not just have a unanimous mechanism in the academy (Nadeau et al., 2010; Edmonds and Mathez, 2017; Le Vaillant et al., 2017...). The current interpretation in the new revision is not friendly with the readers. I strongly suggest that the

preferred reaction/equation of the sulfide liquid oxidation should be clearly pointed out, which is also helpful for the following calculation of the “emanation coefficients”.

2. In the new revision, the authors used the “emanation coefficients” of compound drop system to evaluate element mobility, which borrowed the idea from the arc basaltic volcano metal emissions, in which the emanation coefficient is calculated via the concentrations of element x in the initial magma (C_i) and post-eruptive lava (C_f) like:

$$\epsilon_x = (C_i - C_f) / C_i$$

In the new revision, the authors adopt a similar equation to calculate the emanation coefficients for the compound drop system:

$$[\text{Ec}]_x = (C_{(x,\text{sul})} / d_{\text{sul}} - C_{(x,\text{mag})} / d_{\text{mag}}) / (C_{(x,\text{sul})} / d_{\text{sul}})$$

Here, the authors also introduce the density of magmatic sulfide and replacing magnetite into the equation. I understand the authors’ consideration about this modification, because during the sulfide oxidization, there is mass difference between the magmatic sulfides and replacing magnetite, and we can not directly use the concentrations of elements (i.e., the mass fraction) in sulfide and magnetite for comparison. However, here the density of sulfide/magnetite is put into the denominator, which refers to the volume of sulfide/magnetite per unit mass (e.g., 1 kg or g). Then, in authors’ equation, the concentrations of elements (C_i , sul) should time the volume-like value ($1/d_{\text{sul}}$), and the result becomes a non-physical value.....

In my opinion, the emanation coefficients for the compound drop system should be calculate as the following step:

1) the authors should determine the main reaction form for the sulfide oxidation process, for example:

2) based on this reaction equation, we can obtain that 3 mole FeS ($\sim 3 \times 88$ mole mass) is replaced by 1 mole magnetite (232 mole mass). Hence, the formation of 1 g replacing magnetite should oxidize $\sim 3 \times 88 / 232 \times 1$ g sulfide.

3) Finally, the equation for the emanation coefficients can be written as :

$$[\text{Ec}]_x = (C_{(x,\text{sul})} - 232 / (3 \times 88) \cdot C_{(x,\text{mag})}) / C_{(x,\text{sul})}$$

3. The assumption and concept behind the equation 2 in the new revision are also questionable, and hence the calculated values of magnetite/volatile partition coefficient are not credible.....

In the line 86 (equation 7) in the supplementary discussion, the authors assumed that the sum of the mole fraction of element i in volatile phase and magnetite should equal to 1:

$$X_{\text{sulfide}}^i + X_{\text{(volatile initial)}}^i = 1$$

And then, in the final stage of the compound system, the authors also assume that:

$$X_{\text{magnetite}}^i + X_{\text{(volatile final)}}^i = 1$$

The assumption of these two equations is totally wrong in the thermodynamics sense, and there may be a huge misunderstanding about the mole fraction of component in a multiple system. If the system reach equilibrium, the activity of element i in the magnetite and volatile should be in the same value, but their mole fractions are highly variable due to the mole fraction of other components in magnetite and volatile..... The thermodynamic, basic principle about the sulfide-melt-volatile should be clearly addressed if this work will be submitted in the near future, and the following article may be helpful:

Alain Burgisser, Marina Alletti, Bruno Scaillet, 2015, Simulating the behavior of volatiles belonging to the C-O-H-S system in silicate melts under magmatic conditions with the software D-Compress.....

Besides the wrong assumption mentioned above, the equations 13-16 are also not right. I can not understand the use of molar mass in these equations. I must point out the definition of molar mass of a compound: one mole of any compound will have a mass that is numerically equal to its molecular mass or formula mass and expressed in units of grams. Hence, the molar mass of magnetite (Fe_3O_4) is always about 232 g. If the authors want to know the mole of element i in magnetite, the mole fraction of element i should time to the mole of magnetite (e.g., 4.31 mole for 1 kg magnetite) but not the molar mass of magnetite (~ 232 g)..... In total, the whole calculation for the partition coefficient is questionable, and I try to figure out the right equations for the authors. But this costs me so much time, and may also go out of the scope for a reviewer.....

Minor comments:

Line 105 in the supplementary discussion: should be " $0 < X_{\text{magnetite}}^i < 1$ and $0 < X_{\text{(volatile final)}}^i < 1$ ", but not " $0 > X_{\text{magnetite}}^i < 1$ and $0 > X_{\text{(volatile final)}}^i < 1$ "

REVIEWER COMMENTS MANUSCRIPT NCOMMS-22-45611A

Reviewer #1 (Remarks to the Author):

Manuscript NCOMMS-22-45611-A “Transfer of sulfur and chalcophile metals via sulfide-volatile compound drops in the Christiana-Santorini-Kolumbo volcanic field” by C.G.C Patten, S. Hector, S.P. Kiliyas, M. Ulrich, A. Peillod, A. Beranoaguirre, P. Nomikou, E. Eiche, J. Kolb.

In my previous review I suggested the authors to improve the discussion of two points in order to strengthen the paper:

- The timing of sulfide oxidation and the mechanism operating.

In their reply the authors present a long and comprehensive answer to my comment. I would have preferred that at least part of this answer was incorporated into the main text, it could be useful for other readers.

Yes we agree, it makes sense to add the reply to the manuscript. We have added a paragraph relative to the alternative processes which could have led to sulfide oxidation.

- The calculation and the interpretation of volatile-sulfide partition coefficients.

The text is much clearer now, but I suggest to be even more explicit in explaining (i) what the emanation and partition coefficients indicate (e.g. the order of volatility/mobilization of the investigated elements), and (ii) what the comparison with vapor-melt partition coefficients from the literature indicates (e.g. some elements are more efficiently mobilized by the oxidation of magmatic sulfides than by “normal” magmatic degassing). I would also clarify that the comparison is only qualitative (i.e. relative variations are only discussed, and not calculated E_c and K_d values).

Following reviewers 3 comments, we are not presenting the partition coefficients anymore and the emanation coefficients have been recalculated. We also detail more what these coefficients indicate in the manuscript relative to emanation coefficients from arc volcanoes. Furthermore we discuss the possible genetic link between the magmatic sulfides and the hybrid epithermal-SMS mineralization present at Kolumbo.

Reviewer #2 (Remarks to the Author):

Review of revised manuscript NCOMMS-22-45611A

The authors address my concerns quite well in their responses. However, a few things still need to be

considered:

- In the data provided for individual analysis (chemistry of individual phases - po, ccp, py, mag, and magmatic blebs) there is still in the table a lot of analyses with concentrations in certain elements that are well below the stated LOD for this specific analysis. shouldn't these simply be removed/edited? In particular (but not limited to), this would be important for the following elements and phases as they have an impact on the comparisons and interpretations made in the text): Au in Py and Po and Ti in ovoid magnetite.

We think that data below the limit of detection yield important information and should not be discarded. The very low concentrations of some elements, in magnetite and pyrrhotite, are particularly important for understanding element mobility during sulfide oxidation and calculation of emanation coefficients. This concerns mainly As, Ag, Cd, Sb, Te, Au, Tl and Bi for the magnetite and Ti, Te, W and Au in pyrrhotite. Most of these elements are strongly chalcophile and tend to be enriched in the chalcopyrite. For the calculation of the emanation coefficients we do not use the limit of detection itself as this would lead to an overestimation of the element concentrations but rather half of the limit of detection which is likely closer to the true concentration. In the supplementary table 5 it is stated the values below the limit of detection are reported as half the limit of detection, we realize this information is easy to miss and it is also added in the calculation description. These values are now in italic in the supplementary table 5. We also highlight in Fig. 2 the elements which have values below the limit of detection in magnetite and for which the emanation coefficients are under-estimated.

- Some information given in the response to the reviews need to actually be added in the manuscript itself (in legends - text). Such as

- information on the number of analysed sulfide bleb inclusions, this should be added in the legend of Fig 2.

This is now added in the legend of figure 2. Four sulfide blebs have been analyzed.

- the fact that magnetite compositions presented in Figure 2b are 'magmatic magnetite interpreted as crystallising from the silicate melt

The magnetite compositions in Figure 2b are not magmatic magnetites crystallizing from the silicate melt but the magnetite formed from the oxidation of the sulfides from the compound drops. To avoid confusion it is now detailed in the Figure 2 caption that these are magnetites formed after sulfide oxidation. The magmatic magnetite compositions are shown in supplementary Figure 3.

Also in figure 2b and results presentation, the authors discuss Ni, Co and Mo enrichment of Py and Po compared to ccp. These is really not evident from the data presented in the figure itself.

Yes, indeed, we reformulated the sentence and call the supplementary table 2 where the data are presented.

Apart from the above comments, my main concerns about the manuscript have been answered by the responses to the reviews and edits to the manuscript. Even though I still feel like the story here would be a lot stronger if there was a more statistically representative number of samples/analyses (which does not seem to be possible to increase at this stage without going back on the field and sampling more)

We discuss further the relevance of the number of compound-drops preserved below in reply to reviewer 3 comments.

Reviewer #3 (Remarks to the Author):

This manuscript introduces an interesting sulfide-oxide-vesicles associations occurring in andesitic magmas mixing with dacitic magmas, which was observed in the eruptive products of the Christiana-Santorini-Kolumbo volcanic system. The formation and coalescence of these compound drops can enhance the transfer of sulfur and chalcophile elements in arc magmatic-hydrothermal systems. Meanwhile, the sulfide will be oxidated to magnetite during this process, and the authors also estimate the partition coefficients of chalcophile elements between magnetite and volatile phase. Relative to the original manuscript, some instructive modifications had been made in the new revision under the previous comments and suggestions from the previous reviewers. However, I still think that some important concerns and confusions from the reviewers have not completely/clearly solved in this new revision. In my first review, I pointed out that the calculations of partition coefficients are not right, but the equations in the new revision are still vague or even wrong in the physical meaning, which will be explained in the following major comments. Secondly, the number of sulfideoxide- vesicles associations observed in the CSK volcanic system is highly limited, and the interpretation and claim here, in my opinion, seem to step into an excessive amount of over explaining situation. Based on the current state of this revision, and the limited improvement after the first round of review, my recommendation as of this reading is, rejection. Major comments:

We are thankful, once again, for the extensive and constructive review of reviewer 3. We would like to clarify the concerns of reviewer 3 and 2 relative to the number of observed sulfides and oxides related to compound drops. In our opinion these concerns are understandable but also possibly slightly overstated. We have studied in total 43 samples from Kameni and Kolumbo, with 16 samples having enclaves. Of these 16 samples, 8 have petrographic evidence for sulfide-volatile compound drops, which is far from being anecdotic. We have updated the sample list in the supplementary table 1 and have added additional photos to the supplementary figures 1 and 2.

To understand the apparent scarcity of evidence for sulfide-volatile compound drops it is necessary to consider: 1) how unstable the compound drops are within the dynamic magmatic-hydrothermal system of arc volcano (prone to oxidation, separation of sulfide and volatile phase) and 2) how difficult it is to sample them at the surface in a well preserved state (necessity to transport the mafic-felsic interface at the surface during eruption without too much affecting the compound drops, necessity to find an enclave which can be easily sampled, sulfides can be easily weathered...). Hence evidences for sulfide-volatile compound drops are inherently scarce and when found they are highly valuable. In the landmark paper by Nadeau et al. (2010), the analyzed sulfides blebs are from a total of 12 samples; which one

could also argue to be a limited sample set. Finally, although the number of samples from which evidence of sulfide-volatile compound drops are observed might be limited, there is a high diversity in texture and mineralogy (fresh compound drops preserved in silicate, fresh differentiated ovoids, partially oxidized ovoids and fully oxidized ovoids) and allows for building a comprehensive model of sulfide-volatile compound drop formation and evolution within the CSK volcanic system. Finally similar sulfide-oxide associations, which might be related to sulfide-volatile compound drops, have been reported in the past from different localities such as the Merapedi, Popocatépetl, Pinatubo, Bingham, Tintic and Mount St Helens volcanoes (Nadeau et al., 2010; Larocque et al., 2000). This suggests that preserved sulfide-volatile compound drops might not be limited to the CSK volcanic field and these localities, and others, should be reevaluated.

1. The oxidation process of sulfide is still ambiguous for the readers.

As the point suggested by the reviewer 1, the mechanism of sulfide oxidation is not fully elucidated in the previous version of the manuscript. The authors have made an interpretation for this process within the “Reply to review for manuscript...”, but the in the main part of the new revision, the associated interpretation is still limited and vague. The authors may have considered that they have cite the associated articles in the manuscript, but the sulfide oxidation process does not just have a unanimous mechanism in the academy (Nadeau et al., 2010; Edmonds and Mathez, 2017; Le Vaillant et al., 2017...). The current interpretation in the new revision is not friendly with the readers. I strongly suggest that the preferred reaction/equation of the sulfide liquid oxidation should be clearly pointed out, which is also helpful for the following calculation of the “emanation coefficients”.

Following reviewer 1 we have added an additional paragraph for the possible processes leading to sulfide oxidation. We also add mineral reactions related to sulfide oxidation to help the readers understand the petrology and interpretation.

2. In the new revision, the authors used the “emanation coefficients” of compound drop system to evaluate element mobility, which borrowed the idea from the arc basaltic volcano metal emissions, in which the emanation coefficient is calculated via the concentrations of element x in the initial magma (C_i) and post-eruptive lava (C_f) like:

$$\varepsilon_x = (C_i - C_f)/C_i$$

In the new revision, the authors adopt a similar equation to calculate the emanation coefficients for the compound drop system:

$$Ec_x = \left(\frac{C_{x\ sulf}}{d_{sulf}} - \frac{C_{x\ mag}}{d_{mag}} \right) / \left(\frac{C_{x\ sulf}}{d_{sulf}} \right)$$

Here, the authors also introduce the density of magmatic sulfide and replacing magnetite into the equation. I understand the authors’ consideration about this modification, because during the sulfide oxidization, there is mass difference between the magmatic sulfides and replacing magnetite, and we can not directly use the concentrations of elements (i.e., the mass fraction) in sulfide and magnetite for comparison. However, here the density of sulfide/magnetite is put into the denominator, which refers to the volume of sulfide/magnetite per unit mass (e.g., 1 kg or g). Then, in authors’ equation, the

concentrations of elements (C_i, sul) should time the volume-like value ($1/dsul$), and the result becomes a non-physical value.....

In my opinion, the emanation coefficients for the compound drop system should be calculate as the following step:

1) the authors should determine the main reaction form for the sulfide oxidation process, for example:

2) based on this reaction equation, we can obtain that 3 mole FeS ($\sim 3 \cdot 88$ mole mass) is replaced by 1 mole magnetite (232 mole mass). Hence, the formation of 1 g replacing magnetite should oxidize $\sim 3 \cdot 88 / 232 \cdot 1$ g sulfide.

3) Finally, the equation for the emanation coefficients can be written as:

$$Ec_x = (C_{x \text{ sulf}} - \frac{232}{3 * 88} * C_{x \text{ mag}}) / C_{x \text{ sulf}}$$

We have recalculated the emanation coefficients following the reviewer comments. The new calculations, which takes in account the proportion of pyrrhotite and chalcopyrite in the magmatic sulfides, are detailed in the text and in the supplementary discussion and is now written as:

$$\varepsilon_{x \text{ sulf-mag}} = (C_{x \text{ sulf}} - \frac{M_{\text{mag}}}{3 * M_{\text{sulf}}} * C_{x \text{ mag}}) / C_{x \text{ sulf}}$$

with $C_{x \text{ sulf}}$ and $C_{x \text{ mag}}$ the concentration of the element x in the primary magmatic sulfides and magnetite, respectively and M_{sulf} and M_{mag} the molar mass of the sulfides and magnetite, respectively. The newly calculated emanation coefficients are similar to the ones previously calculated.

3. The assumption and concept behind the equation 2 in the new revision are also questionable, and hence the calculated values of magnetite/volatile partition coefficient are not credible..... In the line 86 (equation 7) in the supplementary discussion, the authors assumed that the sum of the mole fraction of element i in volatile phase and magnetite should equal to 1:

$$X_{\text{sulfide}}^i + X_{\text{volatile initial}}^i = 1$$

And then, in the final stage of the compound system, the authors also assume that:

$$X_{\text{magnetite}}^i + X_{\text{volatile final}}^i = 1$$

The assumption of these two equations is totally wrong in the thermodynamics sense, and there may be a huge misunderstanding about the mole fraction of component in a multiple system. If the system reach equilibrium, the activity of element i in the magnetite and volatile should be in the same value, but their mole fractions are highly variable due to the mole fraction of other components in magnetite and volatile..... The thermodynamic, basic principle about the sulfide-melt-volatile should be clearly addressed if this work will be submitted in the near future, and the following article may be helpful:

Alain Burgisser, Marina Alletti, Bruno Scaillet, 2015, Simulating the behavior of volatiles belonging to the C-O-H-S system in silicate melts under magmatic conditions with the software D-Compress.....

Besides the wrong assumption mentioned above, the equations 13-16 are also not right. I can not understand the use of molar mass in these equations. I must point out the definition of molar mass of a compound: one mole of any compound will have a mass that is numerically equal to its molecular mass or formula mass and expressed in units of grams. Hence, the molar mass of magnetite (Fe₃O₄) is always about 232 g. If the authors want to know the mole of element *i* in magnetite, the mole fraction of element *i* should time to the mole of magnetite (e.g., 4.31 mole for 1 kg magnetite) but not the molar mass of magnetite (~232 g)..... In total, the whole calculation for the partition coefficient is questionable, and I try to figure out the right equations for the authors. But this costs me so much time, and may also go out of the scope for a reviewer.....

We are thankful for the time and effort given by the reviewer to solve the issues related to the calculation of partition coefficients. Following the reviewers comments we come to the conclusion that robust calculations of partition coefficients cannot be done from the available data. The attempt at calculating partition coefficients has been removed. We still want to point out that from the data available the “extraction efficiencies” can be calculated, as defined by Guo and Audetat (2017). These are actually similar to the emanation coefficients except that the dominator is not the initial concentration of the system (i.e. magmatic sulfide) but the final concentration (i.e. magnetite):

$$\varepsilon_x = (C_i - C_f)/C_f$$

Guo and Audetat (2017) highlighted that the extraction efficiency could be qualitatively compared with partition coefficients although it does not have thermodynamic ground. Although we cannot calculate partition coefficients, we think our study could lead the way for further research, from both natural and experimental samples, for determining actual partition coefficients which would be very useful to the community. In replacement of the partition coefficient calculations, we now develop a genetic link between the sulfide-compound drops and the mineralized system of Kolumbo to strengthen the discussion.

Minor comments:

Line 105 in the supplementary discussion: should be “0 < Xmag < 1 and 0 < Xvolatile < 1”, but not “0 > Xmag < 1 and 0 > Xvolatile < 1”

This part of the supplementary discussion has been removed.

REVIEWER COMMENTS

Reviewer #1 (Remarks to the Author):

Manuscript NCOMMS-22-45611-A “Transfer of sulfur and chalcophile metals via sulfide-volatile compound drops in the Christiana-Santorini-Kolumbo volcanic field” by C.G.C Patten, S. Hector, S.P. Kilias, M. Ulrich, A. Peillod, A. Beranoaguirre, P. Nomikou, E. Eiche, J. Kolb.

The authors addressed my concerns about the timing of sulfide oxidation and the possible operating mechanisms: they added a long paragraph dealing with these topic (lines 169-194), which markedly clarify authors’ interpretations.

The authors also addressed my concerns about the calculation and interpretation of volatile-sulfide partition coefficients (also shared by reviewer 3): they removed partition coefficients and recalculated emanation coefficients following the suggestions of reviewer 3; they also clarified the meaning of these emanation coefficients, with respect to those calculated for arc volcanoes (lines 229-239).

I think that the manuscript is significantly improved by these changes and is therefore worth of being published in Nature Communications.

Minor comments

Line 163: “coalesce” instead of “coalescence”

Lines 184-188: probably “fluid” would be clearer than “vol”

Line 186: use “cov” instead of “cv”, to be coherent with the rest of the manuscript

Lines 192-194: the sentence is unclear, please simplify it

Figure 2c: say in the figure legend that emanation coefficients for arc volcanoes are from Edmonds et al 2018.

Reviewer #3 (Remarks to the Author):

Unfortunately, the model calculation of emanation coefficient is still wrong, although I have clearly pointed this in the last three times reviews, which makes me so depressed during this review. In addition, the new-added discussion about the genetic link between compound drop and mineralization is unconvincing and has a wrong logic. In order to show the right calculation equations, my comments in detail had been attached as a pdf document. This revision does not have a good shape to be published.

This new revision has been strongly modified under the previous comments and suggestions from me and other two reviewers. But to be honest, I feel even more disappointed because the calculations/equations in this revision are still wrong, even though the model has been highly simplified in this revision and I had clearly pointed out the authors' errors in my previous two reviews. In addition, this revision deletes the complex calculations of partition coefficients, and the authors new developed a genetic link between the sulfide-compound drops and the mineralized system of Kolumbo to strengthen the discussion. But this part is not new to the researchers, the associated evidence is so weak, and the argument seems to be confusing and ambiguous. The following is my major comments:

1. About the calculation of emanation coefficients:

The authors adopted the form of equation suggested by my last review to calculate the emanation coefficients, which is certainly right. But the authors still misunderstand the "Molar Mass", although I have clearly explained this in my previous reviews. In the Equation S1 within the Supplementary Discussion, the authors should clearly point out that the " X_{Po} " should be the "mass fraction of pyrrhotite", but not just use "fraction" (this can be the mass fraction, volume fraction, or even molar fraction). Because the " X_{Po} " is the mass fraction, the Equation S3 (" $M_{Po} * X_{Po} + M_{Cpy} * X_{Cpy} + M_{SMag} * X_{SMag}$ ") did not calculate the molar mass of magmatic sulfide as declared by the authors, but rather it will obtain the **density** of sulfide because " M_{Po} " is the density of pyrrhotite based on the authors' assumption. Hence, **the Equation S3 here is completely wrong, and the following Equation S4 that is based on S3 to calculate the emanation coefficient is also wrong!!!**

Given that the authors did not understand my interpretation about the molar mass in my last review, I will show to the authors that how to calculate the molar mass of magmatic sulfide:

1. firstly, we can assume that the mass of magmatic sulfide is 1 gram, and hence the mass of pyrrhotite, chalcopyrite and magnetite in this sulfide can be calculated as " X_{Po} " gram, " X_{Cpy} " gram, and " X_{SMag} " gram, respectively;

2. secondly, we can calculate the molar of pyrrhotite, chalcopyrite and magnetite in this magmatic sulfide with mass of 1 gram via that:

Molar of pyrrhotite: $Y_{Po} = X_{Po} / 53.6$ (standard from $-Fe_{1-x}S_x$, ~ 53.6 g per mole if $x=0.1$)

Molar of chalcopyrite: $Y_{Cpx} = X_{Cpx} / 184$ (standard from $-CuFeS_2$, 184 g per mole)

Molar of magnetite: $Y_{SMag} = X_{SMag} / 232$ (standard from $-Fe_3O_4$, 232 g per mole)

3. Thirdly, the sum of molar for this magmatic sulfide with mass of 1 gram can be calculated as:

$$Y_{total} = Y_{Po} + Y_{Cpx} + Y_{SMag}$$

4. Finally, now we know that 1 gram sulfide corresponds to Y_{total} molar, and hence the molar mass of magmatic sulfide (M_{sulf} , following the authors' symbol) means the mass of sulfide when it has 1 mole, which can be calculated as:

$$M_{sulf} = \frac{1}{Y_{total}} = \frac{1}{Y_{Po} + Y_{Cpx} + Y_{SMag}} = \frac{1}{X_{Po}/53.6 + X_{Cpx}/184 + X_{SMag}/232}$$

Because the Equation S3 in the Supplementary Discussion calculate the density of sulfide in fact, the value of this false “ M_{sulf} ” should be around 3500-4500 kg/m³. While the real molar mass of magmatic sulfide should be about ~100-200 (must be larger than pyrrhotite (~53.6 g/mole), and less than magnetite (~232 g/mole)). **The difference between the false and real molar mass of sulfide from the authors and me is larger than one order of magnitude, which will have a huge influence on the calculation of emanation coefficients.** Hence, the authors' mistakes in their calculation can not be neglected, and may strongly change the results.

2. About the genetic link between compound drops and mineralization

This part is firstly added in the new revision to strengthen the discussion, because the calculation model of the partition coefficients had been removed. But this part is not new for the researchers, and James Mungall had clearly demonstrated the genetic link between the compound drops and the ores in Bajo de la Alumbrera porphyry Cu-Au deposit, which was published in 2015. For my viewpoint, relative to the pioneering work from James Mungall, no insightful, new idea has been put forward here, which may be insufficient to strengthen the discussion in the new revision.

Secondly, the listed evidence to support the genetic link between compound drops and mineralization here seems to be unconvincing. The As/Pb ratio of magmatic sulfide has a narrow range (~0.1), while the As/Pb ratio of Kolumbo diffuser can span two orders of magnitude. In additional, the Kolumbo diffuser also has a larger range of Ag/Au ratio relative to that of magmatic sulfide. Although the authors declare that “magmatic sulfide and mineralized samples from the Kolumbo diffusers show good correlation and have similar As/Pb, Bi/Cu and Ag/Cu ratios”, but I do not think the data are convincing. As a comparison,

James Mungall's work shown that the compound drops have nearly constant Cu/Au ($\sim 10^4$) ratio, which perfectly coincides with both ore-forming brines and average orebodies at the Bajo de la Alumbrera porphyry Cu-Au deposit, which clearly confirm the genetic link between the compound drops and the ores. The authors should use some parameters to verify the consistency, not just using the naked eyes.

Thirdly, the most important thing, is that the authors' logic behind this part is chaotic and may be wrong. The authors hope to prove that the oxidation of sulfide-volatile compound drop can directly supply metals to ore deposits (Line 239 and Line 261). Why the authors adopted the primary magmatic sulfides (**before oxidation !!!**) and ore bodies to conduct the compositional comparison. If the data are perfectly matched, the reader can only obtain the information that the metals in ore bodies are from the primary magmatic sulfide, and the sulfide oxidation is not necessary and can be ignorable. Because the As, Pb, Cu and Au have different emanation coefficient, their ratios during sulfide oxidation must be changed. The composition of magnetite is still not suitable for the compositional comparison, because the replaced magnetite represents the remnant of sulfide oxidation, and the major metals had been removed during the oxidation. Hence, in order to confirm the significance of sulfide oxidation process, the authors must compare the compositions of volatile and ore bodies. **This brings the things back to the starting, if the authors can estimate the compositions of volatile, they can directly calculate the partition coefficients, and there is no need to trace the genetic link between compound drops and mineralization to strengthen the discussion.....**

My attitude to this revision is still negative. If the authors adopted my suggestions here, the almost whole model calculations/equations are from me during the three-times' reviews. I do not know how to show/estimate my contribution if this manuscript will be published in the future, especially given that the model calculations are important part of this manuscript. In addition, the new-added discussion about the genetic link between compound drop and mineralization is riddled with holes. This revision is not in a good shape to be published in Nature Communications. **I strongly suggest the authors taking a long time to have a deep thinking and a huge modification before the next submission to this or others journal, rather than just quickly throwing the semi-finished articles to the reviewers.**

Third round of reply to review of manuscript NCOMMS-22-45611B

Reviewer #1:

The authors addressed my concerns about the timing of sulfide oxidation and the possible operating mechanisms: they added a long paragraph dealing with these topic (lines 169-194), which markedly clarify authors' interpretations.

The authors also addressed my concerns about the calculation and interpretation of volatile-sulfide partition coefficients (also shared by reviewer 3): they removed partition coefficients and recalculated emanation coefficients following the suggestions of reviewer 3; they also clarified the meaning of these emanation coefficients, with respect to those calculated for arc volcanoes (lines 229-239).

I think that the manuscript is significantly improved by these changes and is therefore worth of being published in Nature Communications.

Minor comments

Line 163: "coalesce" instead of "coalescence"

Done

Lines 184-188: probably "fluid" would be clearer than "vol"

Done

Line 186: use "cov" instead of "cv", to be coherent with the rest of the manuscript

Done

Lines 192-194: the sentence is unclear, please simplify it

The sentence has been simplified to: "During ISS (chalcopyrite) oxidation, Cu is released into the volatile phase as diverse S- and Cl-complexes⁵³⁻⁵⁵ which equations (4) and (5) are simplified representations."

Figure 2c: say in the figure legend that emanation coefficients for arc volcanoes are from Edmonds et al 2018.

Done

Reviewer #3:

This new revision has been strongly modified under the previous comments and suggestions from me and other two reviewers. But to be honest, I feel even more disappointed because the calculations/equations in this revision are still wrong, even though the model has been highly simplified in this revision and I had clearly pointed out the authors' errors in my previous two reviews. In addition, this revision deletes the complex calculations of partition coefficients, and the authors new developed a

genetic link between the sulfide-compound drops and the mineralized system of Kolumbo to strengthen the discussion. But this part is not new to the researchers, the associated evidence is so weak, and the argument seems to be confusing and ambiguous. The following is my major comments:

1. About the calculation of emanation coefficients:

The authors adopted the form of equation suggested by my last review to calculate the emanation coefficients, which is certainly right. But the authors still misunderstand the “Molar Mass”, although I have clearly explained this in my previous reviews. In the Equation S1 within the Supplementary Discussion, the authors should clearly point out that the “X_{Po}” should be the “mass fraction of pyrrhotite”, but not just use “fraction” (this can be the mass fraction, volume fraction, or even molar fraction). Because the “X_{Po}” is the mass fraction, the Equation S3 (“ $M_{Po} * X_{Po} + M_{Cpy} * X_{Cpy} + M_{SMag} * X_{SMag}$ ”) did not calculate the molar mass of magmatic sulfide as declared by the authors, but rather it will obtain the density of sulfide because “M_{Po}” is the density of pyrrhotite based on the authors’ assumption. Hence, the Equation S3 here is completely wrong, and the following Equation S4 that is based on S3 to calculate the emanation coefficient is also wrong!!!

The confusion relative to equation S3 comes from a typing mistake we made and where we wrote “density” instead of “molar mass” in line 39 of the supplementary discussion (“with M_{Po}, M_{Cpy} and M_{SMag} the density of pyrrhotite, chalcopyrite and magnetite...”). In the S3 equation we did use the molar mass and not the density, which is hinted by the used of “M” and not “d” in the equation. This is also hinted later in the supplementary discussion in line 44 (...the C_{xMag} and M_{Mag} are the element concentration and the **molar mass** of the magnetite...) and 46 (C_{xSMag}, M_{SMag} and X_{SMag} are the element concentration, the **molar mass** and the fraction of the magnetite...). We have corrected the text and we are sorry for the confusion.

Given that the authors did not understand my interpretation about the molar mass in my last review, I will show to the authors that how to calculate the molar mass of magmatic sulfide:

1. firstly, we can assume that the mass of magmatic sulfide is 1 gram, and hence the mass of pyrrhotite, chalcopyrite and magnetite in this sulfide can be calculated as “X_{Po}” gram, “X_{Cpy}” gram, and “X_{SMag}” gram, respectively;

2. secondly, we can calculate the molar of pyrrhotite, chalcopyrite and magnetite in this magmatic sulfide with mass of 1 gram via that:

Molar of pyrrhotite: $Y_{Po} = X_{Po} / 53.6$ (standard from – Fe_{1-x}S_x, ~53.6 g per mole if x=0.1)

Molar of chalcopyrite: $Y_{Cpx} = X_{Cpx} / 184$ (standard from – CuFeS₂, 184 g per mole)

Molar of magnetite: $Y_{SMag} = X_{SMag} / 232$ (standard from – Fe₃O₄, 232 g per mole)

3. Thirdly, the sum of molar for this magmatic sulfide with mass of 1 gram can be calculated as:

$$Y_{total} = Y_{Po} + Y_{Cpx} + Y_{SMag}$$

We agree with the reviewer calculations, which are similar to the one we previously made (using the molar masses and not the densities). The only difference is that we used a different molar mass for pyrrhotite (85.1 g/mole instead of 53.6 g/mole) using the mineral formula of Fe_(1-x)S instead of Fe_(1-x)S_x as suggested by the reviewer and which might be wrong.

4. Finally, now we know that 1 gram sulfide corresponds to Y_{total} molar, and hence the molar mass of magmatic sulfide (M_{sulf} , following the authors' symbol) means the mass of sulfide when it has 1 mole, which can be calculated as:

$$M_{sulf} = \frac{1}{Y_{tot}} = \frac{1}{Y_{po} + Y_{cpy} + Y_{smag}} = \frac{1}{X_{po}/53.6 + X_{cpy}/184 + X_{smag}/232}$$

Because the Equation S3 in the Supplementary Discussion calculate the density of sulfide in fact, the value of this false “ M_{sulf} ” should be around 3500-4500 kg/m³. While the real molar mass of magmatic sulfide should be about ~100-200 (must be larger than pyrrhotite (~53.6 g/mole), and less than magnetite (~232 g/mole)). The difference between the false and real molar mass of sulfide from the authors and me is larger than one order of magnitude, which will have a huge influence on the calculation of emanation coefficients. Hence, the authors' mistakes in their calculation cannot be neglected, and may strongly change the results.

We provide here a table of the magmatic sulfide molar masses from our previous calculations and from the ones suggested by reviewer 3 (but with a molar mass of 85.1 g/mole for Po instead of 53.6 g/mole).

	Mass fraction			Previously calculated molar masses	Newly calculated molar masses from reviewer 3 equations	Difference (%)
Magmatic Sulfide	Po	Cpy	Mag			
Po-Cpy-Mag(85-10-05)	0.85	0.1	0.05	102.3	93.0	-9.0
Po-Cpy-Mag(75-20-05)	0.75	0.2	0.05	112.1	98.8	-11.8
Po-Cpy-Mag(65-30-05)	0.65	0.3	0.05	122.0	105.4	-13.6
Po-Cpy-Mag(55-40-05)	0.55	0.4	0.05	131.8	112.9	-14.3
Molar mass	g.mol ⁻¹					
Pyrrhotite	85.1					
Chalcopyrite	183.5					
Magnetite	231.5					

The differences in the molar masses between the two calculations are below 15% and it does not affect significantly the calculations of the emanation coefficients (see figure below) neither their interpretation.

Over all we think there was a misunderstanding because of a typing mistake we made but ultimately there is little differences in the calculations from our previous version and the newly suggested ones by reviewer 3. To acknowledge the work and commitment of reviewer 3, however, we are using the newly provided equations although it makes little differences relative to the previous manuscript version. The supplementary discussion has been accordingly modified.

2. About the genetic link between compound drops and mineralization

This part is firstly added in the new revision to strengthen the discussion, because the calculation model of the partition coefficients had been removed. But this part is not new for the researchers, and James Mungall had clearly demonstrated the genetic link between the compound drops and the ores in Bajo de la Alumbrera porphyry Cu-Au deposit, which was published in 2015. For my viewpoint, relative to the pioneering work from James Mungall, no insightful, new idea has been put forward here, which may be insufficient to strengthen the discussion in the new revision.

Yes we are aware that the link between compound drops and porphyry deposit was already demonstrated by Mungall et al. 2015. And it is specifically for this reason that we also have to test if the compound drops present in the CSK volcanic field can be involved in the formation of the actively forming hybrid epithermal-VMS mineralization at Kolumbo. By doing so it allows to show that the role of compound drops in the formation of ore deposit is not restricted porphyry and magmatic Ni-Cu-PGE ore deposits but can be also extended to epithermal/VMS deposits, which is a novelty. Additionally, we try to establish the genetic link by using diverse trace element compositions (As, Pb, Bi, Cu, Au, Ag, Tl, Te) from both compound drops and mineralization present in natural samples, which is novel as well. A sentence has been added at the end of the manuscript.

Secondly, the listed evidence to support the genetic link between compound drops and mineralization here seems to be unconvincing. The As/Pb ratio of magmatic sulfide has a narrow range (~ 0.1), while the As/Pb ratio of Kolumbo diffuser can span two orders of magnitude. In addition, the Kolumbo diffuser also has a larger range of Ag/Au ratio relative to that of magmatic sulfide. Although the authors declare that “magmatic sulfide and mineralized samples from the Kolumbo diffusers show good correlation and have similar As/Pb, Bi/Cu and Ag/Cu ratios”, but I do not think the data are convincing. As a comparison, James Mungall’s work shown that the compound drops have nearly constant Cu/Au ($\sim 10^4$) ratio, which perfectly coincides with both ore-forming brines and average orebodies at the Bajo de la Alumbrera porphyry Cu-Au deposit, which clearly confirm the genetic link between the compound drops and the ores. The authors should use some parameters to verify the consistency, not just using the naked eyes.

Yes we agree that the compound drops determined by Mungall have nearly constant Cu/Au ratio which coincides with the **average** ore forming brines and the **average** ore composition. Interestingly, as stated by Mungall et al. (2015), the compound drop have constant Cu/Au because “the metal budget of the compound drop as a whole changes very little because metal is transferred almost quantitatively from the sulphide melt to the vapour”. This is precisely what we observed based on the petrography and on the calculation of emanation coefficients (up to 99% for Cu and 86% for Au, which is a minimum estimate for Au). Hence, there is most likely negligible fractionation between Cu and Au during magmatic sulfide oxidation in the CSK system and the vapor phase should have similar Cu/Au ratio; as modelled in Mungall et al. (2015) study. During sulfide precipitation within the Kolumbo hydrothermal field, however, Cu and Au most likely strongly fractionates from each other, most likely because of temperature of precipitation, as shown by the relatively low hydrothermal fluid temperature (~ 250 °C) at the diffusers. Therefore we cannot use this ratio to establish a genetic link between compound drops and mineralization (the diffusers) at Kolumbo.

More importantly Mungall et al.(2015) are not using actual sulfide composition for their modeling: “The initial metal balance was set to be at equilibrium with **sulphide melt similar** to those reported from sulphide melt inclusions at Alumbrera” (See their supplementary information). These sulfide melt inclusions, from Halter et al. (2005), show wide range of Cu/Au over three orders of magnitude (Fig.3 in

Mungall et al. 2015, see below in red). Also the Cu/Au of the ore fluid inclusions at Alumbrera, from Ulrich et al. (1999), span over 2 to 3 order of magnitudes. Similarly, Nadeau et al. (2010), who linked magmatic sulfides (interpreted by Mungall et al. to be related to compounds) and volcanic gas condensate using various metal ratios such as Fe/Cu, Co/Cu and Ni/Cu also have strong variability in the volcanic gases (2 to 3 order of magnitudes; Fig. 3 in Nadeau et al. 2010, see below in red). Our point here is to highlight that natural samples inherently have high variability, especially in chimneys like in Kolumbo where metal remobilization can occur. Despite the variability in our study the metal ratio range of the magmatic sulfides overlap well with that of the mineralization, especially if we consider only the median value of the mineralization. To make the correlation more evident we have plotted the calculated magmatic sulfide metal ratio range on Fig. 4. To be more conservative and realistic we do not include the Po and Cpy end-members in this range as it is unlikely to have magmatic sulfides with only 100% Po or 100% Cpy (as suggested from the compound inclusions in Sup. Fig. 1). Additionally we have plotted the ranges of “hypothetic vapor phase” metal ratios, which also overlap with the metal ratio ranges of the mineralization (see comments below).

[Figure Redacted]

Thirdly, the most important thing, is that the authors' logic behind this part is chaotic and may be wrong. The authors hope to prove that the oxidation of sulfide-volatile compound drop can directly supply metals to ore deposits (Line 239 and Line 261). Why the authors adopted the primary magmatic sulfides (before oxidation !!!) and ore bodies to conduct the compositional comparison. If the data are perfectly matched, the reader can only obtain the information that the metals in ore bodies are from the primary magmatic sulfide, and the sulfide oxidation is not necessary and can be ignorable. Because the As, Pb, Cu and Au have different emanation coefficient, their ratios during sulfide oxidation must be changed. The composition of magnetite is still not suitable for the compositional comparison, because the replaced magnetite represents the remnant of sulfide oxidation, and the major metals had been removed during the oxidation. Hence, in order to confirm the significance of sulfide oxidation process, the authors must compare the compositions of volatile and ore bodies. This brings the things back to the starting, if the authors can estimate the compositions of volatile, they can directly calculate the partition coefficients, and there is no need to trace the genetic link between compound drops and mineralization to strengthen the discussion.....

Here we disagree with the comments from reviewer 3 and we reply point by point:

“Why the authors adopted the primary magmatic sulfides (before oxidation !!!) and ore bodies to conduct the compositional comparison”

If we had the vapour phase composition we would have used it of course. The composition of the magmatic sulfides is nevertheless useful as the metals selected for the ratios (As, Pb, Ag, Au, Cu, Bi, Tl and Te) have all high emanation coefficients, implying that there is limited fractionation between them during magmatic sulfide oxidation. The metal ratios of the vapor phase should therefore be very similar to those of the magmatic sulfides for the selected metals. This is the same argument brought forward by

the reviewer 3 from Mungall et al. (2015) that the magmatic sulfides and the ore forming fluids have similar metal ratios.

“If the data are perfectly matched, the reader can only obtain the information that the metals in ore bodies are from the primary magmatic sulfide, and the sulfide oxidation is not necessary and can be ignorable”

No, if the data (we assume here data refers to metal ratios) are perfectly matched it means that there is no metal fractionation during metal transfer from the magmatic chamber to the chimneys. Obviously we do not want to suggest to the readers that the magmatic sulfides are physically transported to the chimney...

“Because the As, Pb, Cu and Au have different emanation coefficient, their ratios during sulfide oxidation must be changed.”

Well yes, As, Pb, Ag, Au, Cu, Tl, Te and Bi do have different emanation coefficients (see supp. Table 3), but they are all very high (over 90% for Pb, Ag, Cu, Tl and Bi and up to ~80 % for Au, Te and As; which are likely also higher due to concentration in the magnetite close/at the limit of detection). We did not emphasize that in the previous manuscript, and we apologize, but sulfide oxidation is not a process which fractionates these metals significantly. Hence the vapor phase will have most likely have similar metal ratio as the magmatic sulfides, as shown by Mungall et al. (2015) for the Cu and Au.

We show that now in Figure 4 where we plot the “hypothetic vapor phase metal ratio range” which is defined as the calculated magmatic sulfide ratio (e.g. As/Pb for 10% and 40% cpy) factorized by the emanation coefficients of the selected elements ($E_{C_{As}}/E_{C_{Pb}}$ for 10% and 40% cpy). The ratios for 20 % and 30% cpy falls in-between the 10% and 40% but are not plot for clarity. There are little differences between the magmatic sulfides, hypothetic vapor phases and mineralization ratios, supporting a genetic link. This is now detailed in Figure 4 caption.

“The composition of magnetite is still not suitable for the compositional comparison, because the replaced magnetite represents the remnant of sulfide oxidation, and the major metals had been removed during the oxidation.”

Yes we agree, and this is why we are not using magnetite composition to establish a link with the mineralization.

“Hence, in order to confirm the significance of sulfide oxidation process, the authors must compare the compositions of volatile and ore bodies. This brings the things back to the starting, if the authors can estimate the compositions of volatile, they can directly calculate the partition coefficients, and there is no need to trace the genetic link between compound drops and mineralization to strengthen the discussion.....”

As detailed above, because there is likely minimum fractionation between As, Pb, Ag, Au, Cu, Te, Tl and Bi during sulfide oxidation the metal ratio of the magmatic sulfides can be used as a proxy for the metal ratios of the vapor phase. It is therefore not necessary to actually have the vapor phase concentration to establish a link between compound drops and mineralization.

We have detailed this argumentation in the manuscript from line 263 to 276.

My attitude to this revision is still negative. If the authors adopted my suggestions here, the almost whole model calculations/equations are from me during the three-times' reviews. I do not know how to show/estimate my contribution if this manuscript will be published in the future, especially given that the model calculations are important part of this manuscript. In addition, the new-added discussion about the genetic link between compound drop and mineralization is riddled with holes. This revision is not in a good shape to be published in Nature Communications. I strongly suggest the authors taking a long time to have a deep thinking and a huge modification before the next submission to this or others journal, rather than just quickly throwing the semi-finished articles to the reviewers.

We are really grateful for the reviewer inputs throughout the whole manuscript review process, especially for the calculations. Regarding the contributions of the reviewer, we would argue that the reviewers' main contribution is to have shown us that it was impossible to calculate partition coefficients (which personally I thought it was possible). The calculations are an important part of the manuscript but they are not the core part of it. We would happily acknowledge the time and effort provided by the reviewer in this manuscript, unfortunately he/she is anonymous...

REVIEWERS' COMMENTS

Reviewer #4 (Remarks to the Author):

Dear Editor

I read a ms "Transfer of sulfur and chalcophile metals via sulfide-volatile compound drops in the Christiana-Santorini-Kolumbo volcanic field" by Patten et al. submitted to your journal as well as accompanying reviews-replies of the 3rd circle of revision. As I understood from the conversations, an idea of an original ms was to calculate the metal balance between sulfides and replacing magnetite in volcanics assuming that missing elements enrich volatile phase. In the recent state, I cannot see any shortages in calculation of the emanation coefficients, the approach is ok and the following correlations are valid taking into account natural heterogeneity of the sulfide-poor andesite-dacite systems. It looks like that it was a typing error in the text but the initial calculations were valid.

The genetic link between sulfide-volatile compounds and Kolumbo mineralization is a newly added part of the manuscript and sounds hypothetical; however, indeed, some regularity is seen in the plots in Fig. 4. The authors chose elements, which emanation coefficients are similarly high, to compare their ratios in magmatic sulfides, vapor phase and precipitated mineralization. It is concluded that in the whole magmatic-hydrothermal column (Fig. 3) the selective metal ratios are controlled by the metal proportions in the magmatic source. It is well known that vapor (gas) transport is an effective way of element fractionation (on which gas chromatography is based), however, some elements with similar solid-gas partition coefficients behave similarly and won't fractionate from each other. In general, the thematic of vapor transport is very wide and, inevitably, many assumptions in the ms may require more careful and detailed consideration in a larger paper. For a short paper, the major provocative idea on release of metals due to sulfide oxidation can be accompanied by a hypothetical application to provide a bigger picture. I believe that the authors have the right to suggest this link that is supported by the similarity of the magma sources, the proximity of the volcanic systems and the geochemical constraints. My recommendation is that the ms can be accepted for publication. I have two minor comments, which I leave to the Editor's discretion and author's good will:

I would not recommend using Cu in element ratios as it is a major unevenly distributed constituent of sulfide droplets and blebs, although this is up to the authors.

The model in the Fig.3 does not show position of the Kolumbo diffusers and mineralization. Also, every sulfide droplet is accompanied by a bubble in the cartoon whereas the amount of single-phase vapor bubbles in andesitic melt should be significantly higher than the amount of sulfide-vapor compound droplets.

Technical remarks were not my aims, however, I could not pass by:

Line 258 Merapi volcano

Figure 1i sulfide on photo does not look like pyrite

Replies to reviewer's comments

REVIEWERS' COMMENTS

Reviewer #4 (Remarks to the Author):

Dear Editor

I read a ms "Transfer of sulfur and chalcophile metals via sulfide-volatile compound drops in the Christiana-Santorini-Kolumbo volcanic field" by Patten et al. submitted to your journal as well as accompanying reviews-replies of the 3rd circle of revision. As I understood from the conversations, an idea of an original ms was to calculate the metal balance between sulfides and replacing magnetite in volcanics assuming that missing elements enrich volatile phase. In the recent state, I cannot see any shortages in calculation of the emanation coefficients, the approach is ok and the following correlations are valid taking into account natural heterogeneity of the sulfide-poor andesite-dacite systems. It looks like that it was a typing error in the text but the initial calculations were valid.

The genetic link between sulfide-volatile compounds and Kolumbo mineralization is a newly added part of the manuscript and sounds hypothetical; however, indeed, some regularity is seen in the plots in Fig. 4. The authors chose elements, which emanation coefficients are similarly high, to compare their ratios in magmatic sulfides, vapor phase and precipitated mineralization. It is concluded that in the whole magmatic-hydrothermal column (Fig. 3) the selective metal ratios are controlled by the metal proportions in the magmatic source. It is well known that vapor (gas) transport is an effective way of element fractionation (on which gas chromatography is based), however, some elements with similar solid-gas partition coefficients behave similarly and won't fractionate from each other. In general, the thematic of vapor transport is very wide and, inevitably, many assumptions in the ms may require more careful and detailed consideration in a larger paper. For a short paper, the major provocative idea on release of metals due to sulfide oxidation can be accompanied by a hypothetical application to provide a bigger picture. I believe that the authors have the right to suggest this link that is supported by the similarity of the magma sources, the proximity of the volcanic systems and the geochemical constraints.

My recommendation is that the ms can be accepted for publication. I have two minor comments, which I leave to the Editor's discretion and author's good will:

I would not recommend using Cu in element ratios as it is a major unevenly distributed constituent of sulfide droplets and blebs, although this is up to the authors.

The reviewer raises an interesting point. The magmatic sulfides in the sulfide ovoids are well differentiated, as observed petrographically, and the Cu-rich phase (ISS or chalcopyrite) appears to be preferentially oxidized relative to MSS (or pyrrhotite). The reason for that would be that the Cu-rich phase, being the last one to crystallize during sulfide differentiation would be "on top" at the interphase with the volatile phase of the compounds, and, hence, would be the first phase of the magmatic sulfide to be oxidized (e.g. Fig. 1a and d). This would promote Cu-release during oxidation, despite not being evenly distributed within the magmatic sulfides. Furthermore, the elements used with Cu in the ratios (Bi and Te), show similar distribution to Cu within the magmatic sulfides as they have high partition coefficients into the ISS (e.g. Lui and Brenan, 2015). Therefore, although these elements are unevenly distributed within the magmatic sulfides (between the MSS and ISS) their respective ratios (Cu/Bi or Cu/Te) are most likely not significantly affected depending if MSS or ISS is being oxidized. This is an interesting hypothesis to further study.

The model in the Fig.3 does not show position of the Kolumbo diffusers and mineralization. Also, every sulfide droplet is accompanied by a bubble in the cartoon whereas the amount of single-phase vapor bubbles in andesitic melt should be significantly higher than the amount of sulfide-vapor compound droplets.

The crustal architecture in Fig.3 was based on the Kameni volcano, but it does make more sense to use the one of Kolumbo. We have modified it accordingly. We also have added more vapor bubbles.

Technical remarks were not my aims, however, I could not pass by:

Line 258 Merapi volcano

Corrected

Figure 1i sulfide on photo does not look like pyrite

Yes this pyrite does not look like an typical pyrite, the reason being that it is an oxidized pyrrhotite. We observe similar texture in oxidized, and partially leached, magmatic sulfide in gabbros from the IOPD Hole 735B in the Indian Ocean where pyrrhotite is replaced by pyrite. Of course these are completely different systems but the mineralogical similarities remains and help to understand the textures observed at Kolumbo.